# Caciocavallo Podolico Cheese, a Traditional Agri-Food Product of the Region of Basilicata, Italy: Comparison of the Cheese’s Nutritional, Health and Organoleptic Properties at 6 and 12 Months of Ripening, and Its Digital Communication

**DOI:** 10.3390/foods12234339

**Published:** 2023-12-01

**Authors:** Adriana Di Trana, Emilio Sabia, Ambra Rita Di Rosa, Margherita Addis, Mara Bellati, Vincenzo Russo, Alessio Silvio Dedola, Vincenzo Chiofalo, Salvatore Claps, Paola Di Gregorio, Ada Braghieri

**Affiliations:** 1School of Agricultural, Forestry, Food and Environmental Sciences (SAFE), University of Basilicata, 85100 Potenza, Italy; emilio.sabia@unibas.it (E.S.); paola.digregorio@unibas.it (P.D.G.); ada.braghieri@unibas.it (A.B.); 2Department of Veterinary Sciences, University of Messina, Viale Palatucci 13, 98168 Messina, Italy; ambra.dirosa@unime.it (A.R.D.R.); vincenzo.chiofalo@unime.it (V.C.); 3AGRIS Agris Sardegna, Loc. Bonassai, 07040 Olmedo, Italy; maddis@agrisricerca.it (M.A.); adedola@agrisricerca.it (A.S.D.); 4Behavior and Brain Lab IULM, Center of Research on Neuromarketing, IULM University, 20143 Milano, Italy; bellatimara@gmail.com; 5Department of Business, Law, Economics and Consumer Behaviour “Carlo A. Ricciardi”, IULM University, 20143 Milano, Italy; vincenzo.russo@iulm.it; 6CREA Research Centre for Animal Production and Aquaculture, 85051 Bella, Italy; salvatore.claps@crea.gov.it

**Keywords:** typicality, TAP stretched cheese, cheese fatty acid, retinol, α-tocopherol, cholesterol, polyphenols, antioxidant capacity, nutritional indexes, organoleptic fingerprint, communication and trend on the web

## Abstract

Traditional agri-food products (TAPs) are closely linked to the peculiarities of the territory of origin and are strategic tools for preserving culture and traditions; nutritional and organoleptic peculiarities also differentiate these products on the market. One such product is Caciocavallo Podolico Lucano (CPL), a stretched curd cheese made exclusively from raw milk from Podolian cows, reared under extensive conditions. The objective of this study was to characterise CPL and evaluate the effects of ripening (6 vs. 12 months) on the quality and organoleptic properties, using the technological “artificial senses” platform, of CPL produced and sold in the region of Basilicata, Italy. Additionally, this study represents the first analysis of cheese-related digital communication and trends online. The study found no significant differences between 6-month- and 12-month-ripened cheese, except for a slight increase in cholesterol levels in the latter. CPL aged for 6 and 12 months is naturally lactose-free, rich in bioactive components, and high in vitamin A and antioxidants and has a low PUFA-n6/n3 ratio. The “artificial sensory profile” was able to discriminate the organoleptic fingerprints of 6-month- and 12-month-ripened cheese. The application of a socio-semiotic methodology enabled us to identify the best drivers to create effective communication for this product. The researchers recommend focusing on creating a certification mark linked to the territory for future protection.

## 1. Introduction

Traditional agri-food products (TAPs or TAFPs) are considered essential components of European culture [1,2]. For over five years, the relationship between a territory and its specific cultural identity and tradition has become a robust motivation for customers to purchase traditional products [3]. Recently, data from the XX ISMEA–Qualivita report [4] have shown increasing demand for local or traditional foods, as these foods are often perceived to be of higher quality [5] and more sustainable [3] and to have a solid cultural identity [6] compared with industrial foods. Furthermore, the limited production area contributes to endowing traditional food products with particular characteristics in the eyes of consumers [7]. Italy is the European country with the highest number of Protected Designation of Origin (PDO), Protected Geographical Indication (PGI), and Traditional Speciality Guaranteed (TSG) agri-food products, among which 56 are cheeses [4]. In 1999, Ministero dell’Agricoltura, della Sovranità Alimentare e delle Foreste (MASAF) defined TAPs with DM No. 350 (issued 08/09/99) [8]. The TAP definition reads “obtained with processing methods, storage and maturation over time consolidated, homogeneous throughout the territory concerned, according to traditional rules, for a period not less than twenty-five years” [8]. Currently, the TAP cheese list of the region of Basilicata (southern Italy) consists of 16 products, representing 3.01% of the national list [9]. However, it has to be considered that TAPs have some potentially limiting peculiarities, such as the production being small, seasonal, and often limited to marginal areas. It occurs in family-owned farms according to manufacturing protocols with various types of criticalities and not always in conformity with regulations [2]; in fact, for the production of some TAPs, there are derogations for their production [9]. In 2013, the European Commission created a report to support local manufacturers. The report recommended a labelling scheme for local agricultural products and direct sales; in fact, a new label could add value to these products beyond direct sales if integrated or linked to other measures [10]. The TAP products category, with its peculiarities, is part of two current and important initiatives, i.e., United Nations Decade of Family Farming (UNDFF) 2019–2028 and the recent National Recovery and Resilience Plan (NRRP). UNDFF objectives are focused on promoting sustainable, resilient, inclusive, and viable food systems to stimulate and implement tools in favour of those who depend on family farming [11]. Among the objectives established in the NRRP (Spoke 7), there is giving marginal areas and their products the possibility and opportunity to become territories and products to be valorised.

There are very few studies focused on TAPs and their characterisation; thus, the need to expand knowledge about TAP cheeses emerges. Recently, the distinctive traits of four Apulian TAP cheeses [12,13], a Sicilian TAP cheese [14,15], and a Sardinian TAP cheese [15] were studied. The present work focuses on Caciocavallo (TAP) Podolico produced in the region of Basilicata and named Caciocavallo Podolico Lucano (CPL). The word Lucano derives from the ancient name, Lucania, of the Basilicata region. It is a traditional cheese with Slow Food praesidium (https://www.fondazioneslowfood.com/it/, accessed on 21 July 2023) that represents cultural heritage and is the result of knowledge accumulated and transmitted from generation to generation. CPL, a stretched curd cheese, is produced exclusively with raw milk from Podolian cows. The uniqueness of this cheese arises from the combination and interaction of many factors that contribute to obtaining a distinct product. In fact, its peculiarities may be ascribed to the native cattle breed, the floristic composition of pasture grazed by these animals with its aromatic essences, the pedoclimatic conditions, as well as the cheese-making process, with the wooden tools, length, and conditions of the ripening and anthropic activities being unique and not reproducible elsewhere. Figure 1 and Table 1 report what the cheese looks like and CPL cheese’s general physical [16,17] and organoleptic [18] properties.

Figure 2 reports a flow diagram of the cheese-making process. This traditional cheese has yet to be the subject of in-depth studies aimed at its characterisation. The few studies performed on Podolian milk evaluated its chemical composition [19,20], fatty acid profile, cholesterol [21], and oligosaccharide content [22]. Cheese chemical composition and nitrogenous fractions in relation to the production season [23] and dairy factory [24] were assessed in cheese ripened for 6 and 4 months, respectively. Schena et al. [25] evaluated the lipolysis products during the ripening of CPL, and the differences among factories in total PUFA content were ascribed to different farm pasture diets [24]. Villani et al. [26] provided a general overview of the microbiological aspects of CPL during ripening, while Busetta et al. [24] provided in-depth insights into the microbial populations of CPL during cheese making and ripening, particularly regarding the bacterial biofilms associated with wooden equipment during cheese production. Organoleptic properties have been related to pasture composition [17] and farm differences [24]. To effectively communicate information about CPL cheese, evaluating how product information is currently communicated through web channels is crucial. A valuable approach to achieving this objective is socio-semiotic analysis [27]. This method can help identify the best drivers for communicating information about this cheese effectively.

In light of the above analysis, the objective of this study was to characterize and evaluate the effects of the ripening time (6 vs. 12 months) on the CPL cheese characteristics in order to highlight the peculiar traits of this TAP cheese in terms of gross composition, fatty acid profile, fat-soluble vitamins, cholesterol, total polyphenol content, total antioxidant capacity, nutritional indexes, sensorial fingerprint; also, communication and trends on the web were evaluated.

## 2. Materials and Methods

### 2.1. Survey Design and Cheese Sample Collection

The characterisation of CPL was carried out following the identification, in the region of Basilicata, of all five CPL cheese producers capable of marketing, according to current regulations, the types of ripened cheese most requested by consumers (i.e., with 6 months and 12 months of ripening). On the same day, CPL samples ripened for 6 months and 12 months belonging to all the previously identified manufacturing companies were purchased at cheese shops. For each of the 5 producers, 3 cheeses were sampled for each ripening period. Therefore, the survey’s experimental design involved sampling 30 cheeses (5 producers × 3 cheeses × 2 ripening periods). Each cheese was divided into four aliquots, where one was stored at 4 °C and subjected (within 18 h from collection) to total polyphenol content and antioxidant capacity assays. The second and third aliquots were protected from light and stored at −17 °C to determine gross composition and for artificial sense analysis. The last aliquot was protected from light and held at −20 °C until the fatty acid profile, fat-soluble vitamins, and cholesterol were determined. All analytical assessments for each cheese sample were carried out in duplicate.

### 2.2. Gross Composition

The gross composition of CPL cheese was tested using standard methods. To determine moisture content, 3 g of cheese was analysed using official methods [28]. The fat content was extracted from 3 g of cheese using ether solvents, following the standard procedure [29]. Using the Kjedahl method, the protein content was measured in 1 g of cheese sample [30]. Lactose was assayed in 2 g of cheese according to the standard method [31] and confirmed with Idda et al.’s [32] method. Sodium content was detected in 0.2 g of cheese with nitric acid solution [33,34]. The salt content was then calculated from the sodium content [35]. All measurements were conducted twice.

### 2.3. Fatty Acid Profile

The fatty acid profile of cheese was determined from the extracted fat using the method by Jiang al. [36]. Briefly, 3 g of cheese was mixed with water and isopropanol, and to the mixture, n-hexane was added; then, the mixture was homogenised [15]. The suspension was centrifugated (1094× *g*) at 4 °C for 10 min, and the upper organic layer was transferred into a glass test tube. The n-hexane fraction was extracted and combined with the previous hexane layer. The pooled hexane was evaporated at 30 °C, and the extracted fat was stored at −20 °C until analysis [15]. The fatty acids were methylated using the ISO 15884/FIL 182 method [37]. Fatty acid methyl esters (FAMEs) were analysed with GC-FID using a standard mixture of 37 pure components (Supelco 37 Component FAME Mix; Merck Life Science, Milano, S.r.l., Italy) for identification as reported by Caredda et al. [38]. The isomers of conjugated linoleic acid (CLA) were identified by comparing the retention time of each chromatographic peak with those of a mixture of specific standards (CLA cis9 trans11; CLA trans10 cis12; CLA cis9 cis11; CLA trans9 trans11; Matreya, Restek Italy Super-chrom, Milan, Srl, Italy). A calibration curve with internal standards (100 mg of each per g of fat) was used for the quantitative measurement of each FAME: Me-C5:0 (FAMEs from C4:0 to C6:0), Me-C9:0 (FAMEs from C8:0 to C10:0), Me-C13:0 (FAMEs from C11:0 to C17:0), and Me-C19:0 (FAMEs from C18:0 to C26:0). The classes of FAs were calculated from individual FAs: saturated FAs (SFAs), monounsaturated FAs (MUFAs), polyunsaturated FAs (PUFAs), branched-chain FAs (BCFAs), odd-chain FAs (OCFAs), short-chain FAs (SCFAs), medium-chain FAs (MCFAs), long-chain FAs (LCFAs), omega-6 (PUFA-ω6), and omega-3 (PUFA-ω3). All measurements were performed in duplicate.

### 2.4. Total Retinol, α-Tocopherol, and Cholesterol

The total retinol (vitamin A), α-tocopherol (vitamin E), and total cholesterol content in the cheese samples were determined, in duplicate, using reversed-phase HPLC according to Panfili et al.’s and Manzi et al.’s [39,40] methods. Briefly, to analyse the cheese samples, 0.5 g of cheese was mixed with 2 mL of 60% KOH aqueous solution, 2 mL of 95% ethanol, 1 mL of 1% NaCl aqueous solution, and 5 mL of a 6% ethanolic solution of pyrogallol as an antioxidant. The mixture was digested in a water bath at 70 °C and then cooled for 30 min. To prevent emulsification, 5 mL of 1% NaCl solution was added, and the suspension was extracted with 10 mL of n-hexane/ethyl acetate (9:1, *v*/*v*). The lower aqueous layer was extracted three more times with 5 mL of the same solution. The collected organic layers were evaporated at 30 °C, and the dried sample was dissolved in 3 mL of methanol for HPLC. Finally, a 20 μL sample was injected into the HPLC equipment after filtering the solution using a 0.20 μm PTFE filter.

### 2.5. Total Polyphenol Content and Antioxidant Capacity

Cheese extraction of CPL was carried out according to Rashidinejad et al.’s [41] method with slight modifications, as reported by Di Trana et al. [15]. The extract was stored at −80 °C until further analysis. The total polyphenol content (TPC) in the cheese sample was measured using the Folin–Ciocalteu [42] method in duplicate. The calibration curve was sketched using gallic acid as the reference standard, and the results were expressed in grams of gallic acid equivalents (GAE) per kilogram of cheese. The Ferric Reducing Antioxidant Power (FRAP) assay was conducted twice, following the procedure by Benzie and Strain [43]. A standard curve was created using iron sulphate heptahydrate (FeSO_4_·7H_2_O), and the results were expressed as mmol of FeSO_4_ equivalents per kilogram of cheese. The ABTS Radical Scavenging Activity Assay was carried out twice, following the method described by Re et al. [44]. A standard curve was drawn using a Trolox (6-hydroxy-2,5,7,8-tetramethychroman-2-carboxylic acid) solution, and the results were expressed as mmol of Trolox equivalents (TEAC) per kilogram of cheese. The absorbance readings were taken with a UV-31 UV/VIS spectrophotometer (ONDA; Analytical Instrument, Italy).

### 2.6. Nutritional Indexes

The Health-Promoting Index (HPI) was calculated as suggested by Chen et al. [45]: (n-3 PUFA + n-6 PUFA + MUFA)/[C12:0 + (4 × C14:0) + C16:0]. The General Health Index of Cheese (GHIC), introduced by [46], was modified in light of some meta-analyses on the health components present in dairy products [47,48,49,50]. A new formulation was designed, and the new index (GHIC-7) considers the following seven indicators: butyric acid (C4:0), pentadecanoic acid (C15:0), margaric acid (C17:0), rumenic acid (CLA cis9 trans11), acid α-linolenic acid (C18:3 cis9 cis12 cis15), total antioxidant capacity (TEAC assay), and total polyphenols. For all types of cheese, there are minimum and maximum benchmarks defined for each health indicator. These benchmarks are used to scale the indicators with scores between 0 (indicating low health value) and 10 (indicating high health value). The scores of all the indicators are then added together to obtain the new GHIC-7 index for that type of cheese.

### 2.7. Artificial Sensory Analyses

The sensorial fingerprint of CPL cheese was obtained using an innovative platform of “artificial senses” consisting of an E-eye (IRIS Visual Analyzer VA4000—Alpha M.O.S.), an E-nose (α-Fox 4000—Alpha M.O.S.), and an E-tongue (αAstree—Alpha M.O.S.). Cheese samples of 6- and 12-month-aged cheese were stored at −17 °C; then, they were thawed and left at room temperature for around 15 min before sensory analyses. For the E-eye, each sample was positioned inside the measurement chamber for colour profile analysis to be performed. Sixteen images for each sample were taken against a black background with light from the top [51]. Digital cameras are able to register the colour of any pixel from the image using three-colour sensors per pixel, which capture the intensity of light in the red (R), green (G), or blue (B) spectrum. Chemical compounds showing the five basic taste qualities were evaluated with an E-tongue; for this, 4 g of each sample was minced and made up to 50 mL with bi-distilled water, homogenised for 2 min with Ultra Turrax (T25; Ika Works Inc., Wilmington, NC, USA), and centrifuged at 3000 rpm for 15 min at 4 °C [52]. The obtained solution was filtered and put in a beaker for analysis with seven chemical sensors. Each sample was examined 30 times, and each acquisition lasted 120 s; after each measurement, the sensors were cleaned with bi-distilled water. In order to perform the odour profile, samples were submitted to an E-nose. Samples of 2 g of finely shredded cheese were taken from the middle of each cheese and placed in headspace vials with magnetic caps. For each sample, 4 vials were prepared, incubated for 5 min at 60 °C at agitation speed of 500 rpm for 5 s, and injected [53]. Each acquisition lasted 18 min. Results from the three instruments were used to build the new sensorial fingerprint for 6- and 12-month-ripened cheeses.

### 2.8. Digital Communication and Trends

An analysis was conducted to evaluate the effectiveness of digital communication strategies and consumers’ understanding of CPL cheese. The analysis used a socio-semiotic methodology [27]. The first phase of the methodology was to create a checklist of parameters to assess the impact of communication (focus on the product, rootedness, consistency in narrative, interaction, use of new media, and use of mass media), as well as trends (safety, authenticity, naturalness, sociability, and simplicity). These parameters were chosen based on specific words, adjectives, or other aspects related to cheese. The evaluation of these parameters was conducted using the 1–10 Likert response scale (score 1 = absent; score 10 = fully satisfactory). In the next phase, a test lasting 5 s was carried out to search, on each home page, for each parameter established in the first phase. The parameter values were then recorded, and the average values were plotted in two Kiviat diagrams. As there is no institutional access site for CPL, the first websites that appeared in Google search results were analysed for the study.

### 2.9. Statistical Analyses

Data were tested with the normality test (Shapiro–Wilk) and analysed with Student’s *t*-test using the statistical software Systat SigmaPlot 15 [54]. Results are shown as averages ± SEMs, and differences were considered significant at *p* < 0.05. A tendency was declared at *p* ≤ 0.10. Regarding artificial sense analysis, data from each instrument were subjected to Principal Component Analysis (PCA) for an unsupervised screening of the main lines of variation, in the hope of highlighting any differences between ripening times. Then, the sensors with higher discriminatory power were selected, and the reduced data set was subjected to a further PCA to improve the ability to analyse the results and avoid redundancy in the sensors’ responses. Data evaluation was performed based on the discrimination index (DI), which gives an evaluation of the discrimination quality of the selected plane from the surface between groups and the size of each group. In addition, based on the organoleptic distance, the pattern discrimination index (PDI%) among the 2 groups was calculated. Further, discriminant function analysis (DFA) was employed to conduct the classification task. All data analyses were performed using the native instrument AlphaSoft statistical software v12.44.

## 3. Results and Discussion

### 3.1. Gross Composition of CPL Cheese

Gross composition, particularly moisture, fat, protein, and salt content of cheese, is a complex function related to milk quality and the cheese-making process. Like other autochthonous breeds reared for their ability to adapt to local climatic and territory characteristics, Podolian cows generally produce milk with higher protein and fat contents [19,21] compared with selected breeds due to the lower production level. Generally, high cheese yields require high concentrations of fat and casein, the so-called cheese-making useful matter, in processing milk. The gross composition of CPL did not differ significantly based on the ripening months, as shown in Table 2. The average moisture values ranged from 33 g/100 g to 29 g/100 g of cheese, which is normal due to water loss during cheese ripening. The average values of fat and protein ranged from 30 g/100 g to 32 g/100 g of cheese and 31 g/100 g to 32 g/100 g of cheese at 6 and 12 months of ripening, respectively. The ash and salt contents also showed trends similar to those of other parameters, with average values of 5.9 g/100 g to 6.6 g/100 g of cheese and 2.3 g/100 g to 2.6 g/100 g of cheese, respectively, at 6 and 12 months of ripening. The trends of the gross composition parameter values during ripening may have been due to a concentration effect caused by the cheese losing moisture. Fallico et al. [55], on Ragusano cheese, a pasta filata cheese made from raw milk, observed a negative correlation between moisture and protein content.

Generally, lactose is not completely hydrolysed during the curd acidification phase. The amount remaining in fresh cheese depends on several factors, with the most important ones being the type of starter used in the cheese-making process and the duration of the curd acidification phase [24,32,56]. The residual lactose content is further reduced during cheese ripening; long-ripened cheeses are generally naturally lactose-free [57]. The General Directorate for Hygiene and Food Safety and Nutrition of the Ministry of Health (DGSAN 0024708 of 16 June 2016), based on the opinion expressed by the European Commission for Dietetics and Nutrition (4 May 2016), has given indications that the claim “naturally lactose-free” may be used for cheese with a lactose level of less than 0.1 g in 100 g (or 100 mL) of product. In hard and mature cheeses, during the ripening process, the lactic bacteria consume all the lactose contained in the cheese [58]. The lactose content in hard and long-maturing cheeses is generally very low and can be tolerated by most individuals suffering from primary lactose [59] intolerance. The results obtained in the present study highlighted a significant outcome, namely, CPL is a “naturally lactose-free” cheese (0% for 6- and 12-month-aged cheese). In Table 2, the lactose content was declared equal to 0 g/100 g of cheese, based on the guidance document on the rounding rules applicable to the declaration of nutrients (European Commission Health and Consumers Directorate-General, December 2012: Guidance document for competent authorities for the control of compliance with EU legislation on Regulation (EU) No 1169/2011, Council Directive 90/496/EEC and Directive 2002/46/EC). According to this guidance document, if the sugar content in a food is less than or equal to 0.5 g/100 g, it can be declared as 0 g/100 g in the nutrition table. Given the importance of this declaration, also in light of the fact that lactose intolerance is very widespread and since the official method used for the determination of lactose [31] did not report any Limits of Detection and Quantification, the samples of CPL were also analysed with another validated analytical method [32,33]. This method, which involves the determination of lactose using gas chromatography with a Limit of Detection (LOD) of 0.54 mg/kg of cheese and a Limit of Quantification (LOQ) of 1.68 mg/kg of cheese, confirmed that the lactose content in CPL is, on average, 0.001 g/100 g, thus lower than the limit set by the aforementioned ministerial note; therefore, the cheese can be declared “naturally lactose-free”. This characteristic of CPL represents a further added value of this TAP cheese. As with other cheeses identified as TAPs, the gross composition variability can be attributed to the compositional fluctuation of milk depending on the season, the rearing conditions, and the feeding of the cows, as well as to the lack of standardised procedures for the making and ripening of CPL cheese. Perna et al. [23] observed variations in the gross composition, in terms of moisture, total protein, and fat content, of Caciocavallo Podolico cheeses at the same ripening time during the cheese-making period and sampling (from January to December) as the effect of environmental factors and the quality and quantity of forage intake by Podolian cows at pasture. Pizzillo et al. [17] ascertained a chemical composition variation in Caciocavallo Podolico cheese mainly linked to the production area, rather than the ripening age. Recently, Natrella et al. [60] did not detect significant differences in gross composition between Caciocavallo cheeses aged for 180 and 340 days. The moisture, fat, and protein values of CPL cheese agree with what was reported in the aforementioned study.

### 3.2. Fatty Acid Profile of CPL Cheese

The fatty acid profile of CPL did not differ significantly (*p* ≥ 0.10) based on the ripening months, as shown in Table 3. In both products, twenty-five fatty acids (FAs) from C4:0 to C26:0, including FAs with health functions, such as rumenic acid, the main isomer of CLA (C18:2 cis9 trans11-RU), and alpha-linolenic acid (C18:3 cis9 cis12 cis15-ALA), were identified in quantities > 0.1 g/100 g of cheese. The averages of odd- and branched-chain FAs (ΣOBCFAs; Table 3) showed 1.4 and 1.6 g per 100 g of cheese at 6 and 12 months of ripening, respectively. Currently, OBCFAs, considered biomarkers of human fat intake from dairy products, constitute an emerging class of bioactive compounds that, according to recent studies in humans, induce beneficial effects on metabolism and favourably influence health at cellular and systemic levels [61,62]. Consequently, the total content of branched-chain FAs (∑BCFAs) in cheese assumes an essential role as an exogenous source available to humans. The BCFA content in cheese varies depending on the type of cheese and its fat content [63]. In CPL cheese, which had 30% and 32% fat at 6 and 12 months of ripening, respectively, the averages of BCFAs were 193 mg and 208 mg per serving (assuming a serving size of 28.35 g). These values are higher than the 148 mg per serving of Cheddar cheese, which has 33% fat [63]. The contents of BCFAs and OCFAs in milk and dairy products varies according to the ruminant breed [64], physiological stage [65,66], and diet of the animal [67]. Furthermore, OBCFA content and proportion reflect variations in the rumen’s bacterial population, and many OBCFAs derive from mammary gland de novo synthesis [68]. Based on the information provided above, the Podolian breed and the extensive rearing system play a pivotal role in the OBCFA content in CPL cheese. The obtained result agrees with the OBCFA content found in Caciocavallo Palermitano (CP), a TAP cheese produced with raw milk from the local cattle breed, the Cinisara breed, reared in an extensive farming system [69]. In CPL cheese, as it typically happens in dairy products, the class of short-chain fatty acids (SCFAs; Table 3) was less represented (1.98 and 2.2 g/100 g of cheese) than that of medium-chain fatty acids (MCFAs; 13.5 and 14.7 g/100 g of cheese) and long-chain fatty acids (LCFAs; 13.1 and 13.3 g/100 g of cheese) in cheese at 6 and 12 months of ripening, respectively. Within SCFAs, butyric acid (C4:0) was the primary representative, with average content of 1.06 g/100 g of CPL cheese, which is higher than the content found in CP cheese (3.25 g/100 g FA), a TAP raw milk cheese produced in Sicily [14]. C4:0 content is generally higher in bovine milk than in sheep and goat milk [70] and performs multiple functions in the human organism. For instance, it is enterocytes’ primary energy source and has a central role in maintaining homeostasis and gut health [71]. The class of MCFAs (from C10:0 to C16:1) was the most representative, with 47% and 49% of total FAs in CPL cheese at 6 and 12 months of ripening, respectively (Table 3). In this class, C16:0, with 58% and 56% (in CPL aged for 6 and 12 months, respectively), was the most abundant. Concerning LCFAs (from C17:0 to C26:0), their percentages reached 46% and 44% of total FAs; among LCFAs, C18:1 cis9 was the primary fatty acid (44% and 39% in cheese at 6 and 12 months of ripening, respectively). The percentages of saturated fatty acids (SFAs) in CPL cheese were 66% and 69% in 6- and 12-month-ripened cheese, respectively, with C16:0 being the most prevalent. Monounsaturated FAs (MUFAs) were 29% and 26% of total FAs in cheese at 6 and 12 months of ageing, respectively; in this class, the main fatty acid was oleic acid (C18:1 cis9), representing 69% and 66% of MUFAs, followed by vaccenic acid (C18:1 trans11-VA), representing 7% and 11% of MUFAs, in cheese aged for 6 and 12 months, respectively. Finally, in both 6- and 12-month-old CPL cheese, polyunsaturated fatty acids (PUFAs) made up 4.9% and 5.0% of total FAs, respectively. The most abundant PUFA was linoleic acid (C18:2 cis9 cis12-LA), which accounted for 38% and 28% of the total amount in 6- and 12-month-aged cheeses, respectively. Following linoleic acid, RU (C18:2 cis9 trans11-RU) made up 21% and 24% of PUFAs, and ALA (C18:3 cis9 cis12 cis) made up 14% and 18% of PUFAs in 6- and 12-month-aged cheeses, respectively. Among MUFAs and PUFAs, the VA, RU, and ALA individual FAs were well represented (0.73, 0.33, and 0.24 g/100 g of cheese, respectively), in line with the known link between the use of pasture in extensive farming systems and the increase in these FAs in cheeses [72,73]. As concerns RU acid, similar values were observed in two raw cow milk cheeses produced mainly in extensive farming systems, i.e., CP cheese (0.31 g/100 g of cheese) and CM (Casizolu del Montiferru) cheese (0.19 g/100 g of cheese) [15]. Studies and reviews of in vivo human models, over the last 13 years, on the beneficial effects of these FAs have shown that the ingestion of cheese naturally enriched with RU, VA, and ALA has beneficial properties, as it improves the lipid profile of plasma and, significantly, reduces the biosynthesis of endocannabinoids [74,75,76]. In CPL cheese, as well as in dairy products, the n6/n3 ratio reflects the contents of LA and ALA, and this health index improves by increasing the fresh grass intake by ruminants [72,73]. The mean value of the PUFA-n6/n3 ratio in CLP cheese was 2.2, and this value is close to that of CP cheese (2.0) and lower than that of CM cheese (2.9) [15]. A low value of this ratio mirrors the extensive farming system through which CPL cheese is produced, and it is in line with the Department of Health’s nutritional recommendations for consumers that foods should not exceed 4:1 in terms of PUFA-n6/n3 ratio [77].

### 3.3. Fat-Soluble Vitamins and Cholesterol in CPL Cheese

The contents of the parameters reported on the mandatory nutritional declaration of cheeses (protein, fat, and NaCl) are reported in Table 2; according to EU regulation No. 1169/2011 [35], it can be integrated with non-mandatory parameters, such as some FAs (e.g., MUFAs and PUFAs) and vitamin contents. Table 4 shows CPL cheeses’ total retinol (vitamin A), α-tocopherol (vitamin E), and cholesterol contents. Total retinol and alpha-tocopherol levels in CLP cheese were similar at 6 and 12 months of age (*p* ≥ 0.10). The average values of total retinol were 2.0 mg/kg for 6-month-old cheese and 1.9 mg/kg for 12-month-old cheese. Similarly, the average values of alpha-tocopherol were 9 mg/kg for 6-month-old cheese and 7 mg/kg for 12-month-old cheese. These values are in line with those found in CP (2.5 and 6.6 mg/kg of cheese) and CM (2.6 and 8.6 mg/kg of cheese) cheeses, where both are TAP cheeses produced in extensive systems [15]. The variations in total retinol and alpha-tocopherol in milk and cheese depend mainly on the dietary supply, even during the lipid mobilisation period [78] of animals. In pasture-fed animals, seasonal changes in the availability and quality of fresh grass are the main factors affecting fat-soluble vitamin content in milk and cheese [79]. Revilla et al. [80] observed that the levels of vitamins (retinol and α-tocopherol) in bovine cheese decreased from 0 to 1 month and from 0 to 2 months of ripening, while no significant differences were found between 1 and 2 months of maturation. These fat-soluble vitamins in milk and dairy products are pivotal in preventing human diseases; retinol is essential for the developing and correct functioning of the immune system, while alpha-tocopherol prevents the oxidation of lipids and cholesterol [81]. A portion of 100 g of CPL cheese can provide a quantity of vitamin A equal to 197 µg, and because of this, this cheese can be considered a source of vitamin A, according to EC regulations (Directive 2008/100/EC; Regulation (EC) No. 1924/2006).

Total cholesterol content in CPL cheese slightly increased (*p* = 0.057) from 6 to 12 months of ripening (872 vs. 1089 mg/kg of cheese; Table 4). These levels are close to those found in CP cheese after 2 months of ageing (820 mg/kg of cheese) and in CM cheese aged for 6 months (1061 mg/kg of cheese) [15], as well as in Provolone cheese (750 mg/kg of cheese) and Parmesan cheese (926 mg/kg of cheese) [82]. These parameters contribute to improving the information about the nutritional characteristics of CPL cheese, which is currently scarce or not available, and to updating the Italian Food Composition Database [83].

### 3.4. Polyphenol Content and Total Antioxidant Capacity of CPL Cheese

Table 5 depicts CPL cheese’s total polyphenol content (TPC) and total antioxidant capacity (CAT), measured with FRAP and TEAC assays. No significant difference (*p* > 0.10) in these parameters was observed based on the time of ripening. The results of TPC, which characterises CPL cheese, cannot be directly compared with similar research results because these compounds were measured for the first time in CPL cheese. However, TPCs in two raw cow milk cheeses produced mainly in extensive farming systems in Sicily and Sardinia [15] and in one raw milk Alpine cheese made in summer and winter [84] were recently evaluated. In CPL cheeses ripened for 6 and 12 months, the average values of TPC were (4.1 and 5 g GAE/kg of cheese, respectively) higher than those found in CP cheese produced in the winter (3.52 g GAE/kg of cheese) and spring periods (4.65 g GAE/kg of cheese); in CM cheese obtained in February, May, and September (2.98, 3.25, and 3.65 g GAE/kg of cheese, respectively) [15]; and in Ossolano-like cheese produced in the summer and winter periods (3.27 and 2.88 g GAE/kg of cheese, respectively) [84]. Polyphenols, widely diffused in the plant kingdom, are a vast and complex group of compounds characterised by various beneficial effects for human health [85] due to their antioxidant, antimutagenic, and antitoxic capacities [86]. In extensive and semi-extensive breeding systems, in which natural pastures, rich in phenolic compounds, constitute the primary source of animal feed, polyphenols play a central role in animal digestion and animal performance, as well as in the quality of milk and derived products [87,88]. TPC in milk and cheeses is related to the feeding system; in fact, the pasture management system improves cow milk polyphenol content [89]. Noteworthy is the variability in TPC in the dairy products from grazing animals, which depends on the type of forage species [46,90,91,92], grazing season [93,94,95], aromatic plant intake [96], plant phenological stage, and the biodiversity of the ingested herbs [97,98]. Moreover, the abovementioned factors and polyphenol bioavailability interact with ruminal and intestinal microbiota action [99]. It is known that the bacterial population modulates rumen degradation and intestinal absorption of metabolites produced from dietary phenolic compounds [100,101]. Various authors have suggested that the phenolic composition of milk and cheese could be used to establish a fingerprint to track the animal’s diet [91,102]. In CPL cheese, the FRAP assay and TEAC (ABTS assay) were used to evaluate the CAT (Table 5). The CAT values, measured using the FRAP test, were similar for 6-month and 12-month-ripened CPL cheese (1.4 mmol FeSO_4_/kg of cheese). However, these average values were slightly lower compared with the traditional CM and CP raw milk cheeses from Sardinia and Sicily, where the values ranged from 1.69 to 2.08 mmol FeSO_4_/kg of cheese for 6-month-ripened CM cheese and from 1.84 to 2.00 mmol FeSO_4_/kg of cheese for 2-month-ripened CP cheese [15]. Based on the results in Table 5, there was no significant difference in the antioxidant properties assayed with TEAC between the ripening times. The average TEAC values ranged from 67 mmol Trolox/kg of cheese to 69 mmol Trolox/kg of cheese at 6 and 12 months of ripening, respectively. These results highlight that CPL cheese has higher antioxidant properties compared with CM cheese ripened for 6 months (10.3–18.9 mmol Trolox/kg of cheese) and CP cheese ripened for 2 months (46.8–52.4 mmol Trolox/kg of cheese) [15]. In various cheeses, the CAT increased up to 4 months and then decreased up to 9 months of ripening [80,103,104]; this trend was correlated with the degree of proteolysis, which occurs in cheese during ripening [103], and the decrease observed after 4–5 months of maturation was attributed to antioxidant peptides being unable to resist continuous proteolysis [80]. Antioxidants are chemical compounds that can neutralise and scavenge the free radicals continuously produced in the human body [105]. Intake of antioxidants through foods naturally rich in these compounds may protect the body from oxidative stress and damage [106]. Dairy products contain naturally occurring antioxidant substances in variable proportions depending on various factors: the origin of the raw material; different concentrations of vegetable antioxidants naturally present in the animal’s diet; contents and types of caseins, whey proteins, fat-soluble vitamins and precursors (retinol, α-tocopherol, and β-carotene), uric acid, phenols, folates of microbial origin; cheese-making process; type of coagulant; production season; and ripening time [46,104,107,108,109,110]. An extensive database of 3100 vegetable and non-vegetable foods, including milk and dairy products, was created using the global dosage of CAT [111]. A positive correlation between TPC and TEAC values was obtained for CPL cheese for both ripening times (0.81, *p* < 0.05). This result is in agreement with the same correlations observed by Kuhnen et al. [89] in milk and by Přikryl et al. [112] in cheese. Contextually, some authors have proposed the CAT of the diet as a potential marker of the quality of the diet [113]. The CAT value of cheese is affected by numerous factors and their interactions.; the optimal CAT seems to depend on the type of cheese and the ripening time, combined with the specific microbial activity that occurs during ripening. However, in our study, the ripening factor had no effect on the CAT of CPL cheese.

### 3.5. Nutritional Indexes of CPL Cheese

Nutritional indexes are tools used to evaluate and compare the nutritional value of FAs and, given their potential for the prevention of certain diseases, to obtain useful information about the nutritional and nutraceutical value of food products. The Health-Promoting Index (HPI) of dairy products has values from 0.16 to 0.68 [45]; a high HPI value is assumed to be more beneficial to human health. In Table 6,CPL showed values (0.51 and 0.44 at 6 and 12 months of ripening, respectively) close to the upper limit indicated; these values align with those reported for CP cheese [14]. No significant differences were found between ripening times. The HPI value of 12-month CLP can be explained by the primary biochemical changes that occur during the cheese-ripening process, such as lipolysis [114]. This leads to an increase in AG in mature cheeses, causing the denominator of the HPI index to increase and the index itself to decrease. The new GHIC-7 index combines the healthy compounds (C18:2 cis9 trans11-CLA, antioxidant capacity, and total polyphenols) already used in the calculation of the GHIC index [46] and some health components present in dairy products emerged from meta-analyses [48,49,50,104]; in particular, butyric acid (C4:0), pentadecanoic acid (C15:0), margaric acid (C17:0), and α-linolenic acid (C18:3 cis9 cis12 cis15-ALA) are used. The GHIC-7 score ranged from 32 to 39 in CPL ripened for 6 and 12 months, respectively. It was observed that the ripening process had no effect on GHIC-7. However, humidity levels and antioxidant capacity may have influenced the values. According to Gupta et al. [103], higher maturation time can lead to the formation of soluble peptides, which is positively correlated with antioxidant capacity.

### 3.6. Artificial Sensory Profile of CPL Cheese

Artificial senses were very useful for determining an innovative organoleptic fingerprint for CPL. Figure 3 and Figure 4 show the spider charts of the sensors’ responses to 6- and 12-month-ripened cheeses in terms of colour, odour, and taste profiles provided by an E-eye, an E-nose, and an E-tongue, respectively. In Figure 5, PCA plots of results obtained with the E-nose and E-eye show the high variability within each group, highlighting the colours and volatile class developed, which derive from numerous production factors in addition to ripening. In detail, the E-nose LY-type sensors, sensitive to chloride and short-chain volatile fatty acids, moved toward cheeses aged for 6 months, while those more ripened (12 months) were defined by *p*-sensors, which are sensitive to propane, methane, and other aliphatic nonpolar molecules, and by T-type sensors, which are sensitive to polar alcoholic and chlorinated compounds [115]. The E-eye results show that the RGB colour codes defining the cheeses aged for 12 months were darker (1891, 1892, 2164, 2165) than those of the cheeses aged for 6 months (2437, 2438, 2436, 2454). In other cheese types, such as Palmero PDO [116] and Emmental cheeses [117], an increase in colour intensity with ripening time has already been observed. The evolution in cheese colour is related to changes in protein hydration during ripening, the decrease in the quantity of free moisture, and consequently, the light-scattering characteristics of the cheese matrix [118,119]. The PCA plot obtained from the results of the E-tongue (Figure 4) shows an effective discrimination power between the two different ripening periods. The first two planes (PC1 and PC2) represent 96.2% of the total variance among sample measurements, with a discrimination index of 52%. The samples are clustered in the bi-dimensional space according to the ripening period. This result is probably related to the physical–chemical changes in cheese during ripening [120,121,122]. Moreover, the ripening period and the cheese-making procedure, including the use of starter cultures and ripening conditions, may significantly influence the organoleptic properties of cheese [123,124,125]. The results from organoleptic data fusion (Figure 6), combining colour, smell, and taste, like a human panel, objectively underline the clear difference between the two groups of CPL for two different ripening times.

### 3.7. Communication and Trends Relative to CPL Cheese on the Web

The results of the socio-semiotic evaluation linked to the communication analysis of the CPL web pages are depicted in two graphs, Figure 7 and Figure 8. The major weaknesses identified through the evaluation were internationality (score 2), due to the sites being available only in Italian without the possibility of translation, and interaction (score 2), due to the absence of an interactive community. Poor use and connection with traditional mass media (score 3) and new media, such as social networks (score 3), were also highlighted. However, the evaluation identified several strengths, including a consistent narrative (score 7), well linked to the product and its origins, and a strong focus on the product (score 8), which is central to communication. The evaluation checklist also showed that the communication trends (Figure 8), including authenticity, simplicity, security (score 7), and naturalness (score 8), were effectively transmitted via web communication. The only aspect lacking was the social aspect (score 5) due to the lack of a community and connection with new media, which limits interaction. These results agree with Philippidis et al. [126], who observed that the concept of authenticity, associated with traditional cheese, is considered one of the main drivers in consumers’ attitudes towards brands and products [127]. The perception of territoriality leads consumers to a greater propensity to buy and a greater sense of perceived safety and emotion [128]. Indeed, the safety perception was shown to play a mediating role between the perception of territoriality and the propensity to buy [129]. Additionally, van Ittersum et al. [130] confirmed the fundamental importance of the certification labels as a symbol of protection connected to the territory. A strong point for CPL cheese is the presence of salient information, especially about the attention to the narrative in storytelling, which very well emphasises the strengths linked to authenticity and territoriality, and the presence of producers and specialised food and wine sites. The use of this socio-semiotic analysis can be considered a starting point for a critical evaluation of the state of the art of web communication in the traditional dairy sector of southern Italy. This analysis fits into a broader and more specific context for the preparation and optimisation of drivers useful for effective CPL product communication.

## 4. Conclusions

The results revealed that there were no significant differences between cheese samples at 6 and 12 months of ripening, except for a slight increase in cholesterol levels in 12-month-old cheese. It was highlighted, for the first time, that CPL cheese is naturally lactose-free and contains higher levels of bioactive components, such as BCFAs, compared with other cheeses with similar fat content. Its rumenic acid content is comparable to that in cheese from raw milk from animals raised in extensive systems. It contains an excellent source of vitamin A and polyphenols, and good nutritional value, with a low PUFA-n6/n3 ratio, in line with nutritional recommendations. Both cheeses, ripened for 6 and 12 months, are also suitable for the class of consumers who are lactose intolerant. The innovative technology with artificial sensory profile could characterise and discriminate between the 6-month-old and 12-month-old cheese samples. The socio-semiotic analysis applied to CPL identified useful drivers for preparing and optimising effective CPL product communication. The researchers suggest creating a certification mark to protect the product and identify the territory and establishing a producers’ cooperative as future objectives.

## Figures and Tables

**Figure 1 foods-12-04339-f001:**
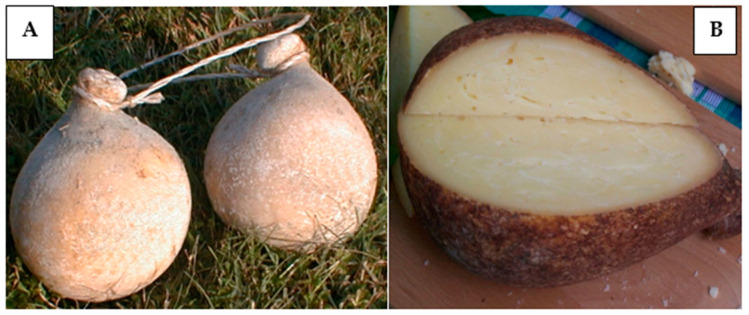
Pair of whole Caciocavallo Podolico Lucano cheeses ripened for 12 months (**A**) and cheese cut open (**B**).

**Figure 2 foods-12-04339-f002:**
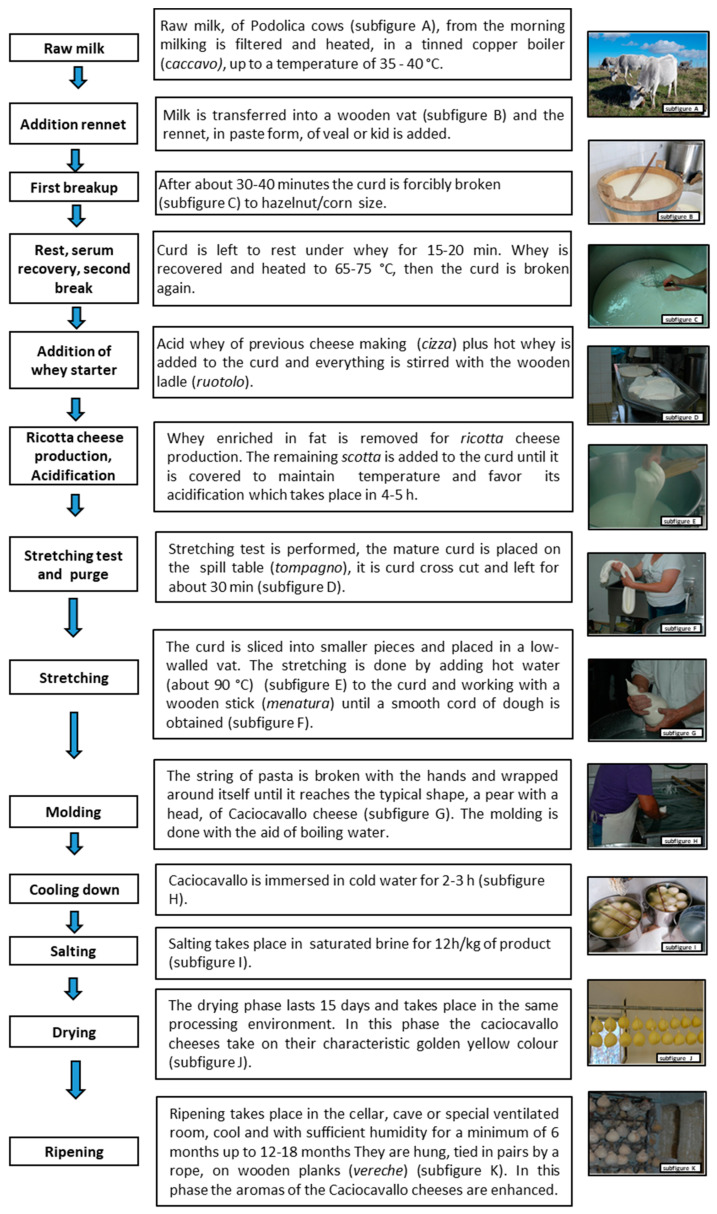
Flow diagram of Caciocavallo Podolico Lucano cheese-making process (photos by Lovallo, C.).

**Figure 3 foods-12-04339-f003:**
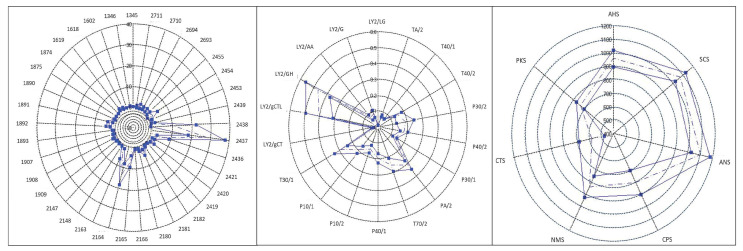
Organoleptic fingerprints of CPL cheese at 6 months obtained with E-eye, E-nose, and E-tongue.

**Figure 4 foods-12-04339-f004:**
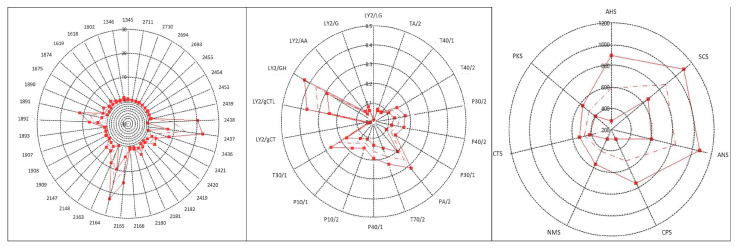
Organoleptic fingerprints of CPL cheese of 12 months of age obtained with E-eye, E-nose, and E-tongue.

**Figure 5 foods-12-04339-f005:**
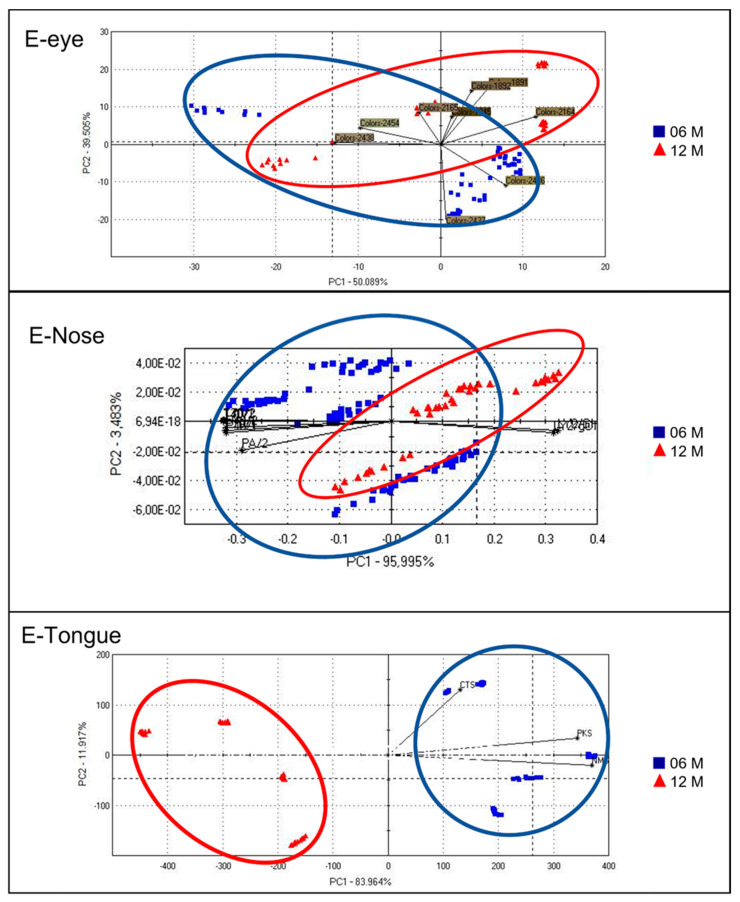
PCA plot of E-eye, E-nose, and E-tongue results of Caciocavallo Podolico Lucano, grouped according to the ripening period (6 months, blue; 12 months, red). DI= −8 for E-eye; Di= −9 for E-nose; DI= 52 for E-tongue.

**Figure 6 foods-12-04339-f006:**
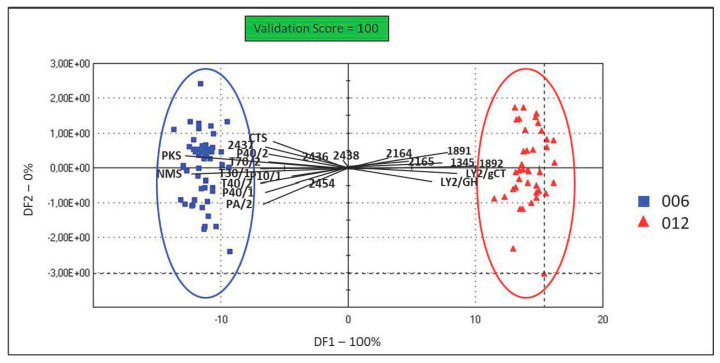
DFA plot of sensor fusion of Caciocavallo Podolico Lucano grouped according to the ripening period (6 months, blue; 12 months, red). Validation score = 100.

**Figure 7 foods-12-04339-f007:**
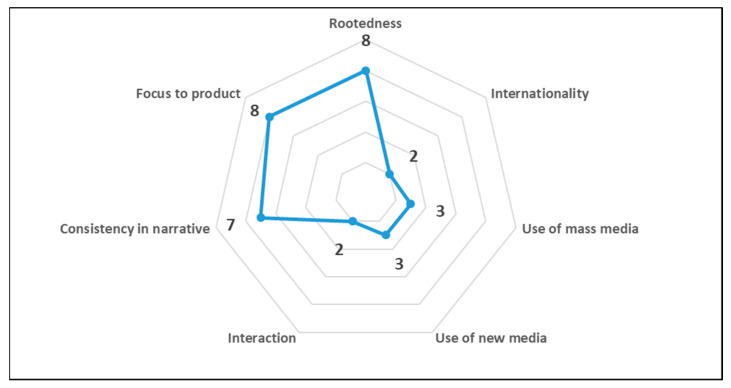
Assets of CPL cheese communication (scale: 1 = absent; 10 = fully satisfying).

**Figure 8 foods-12-04339-f008:**
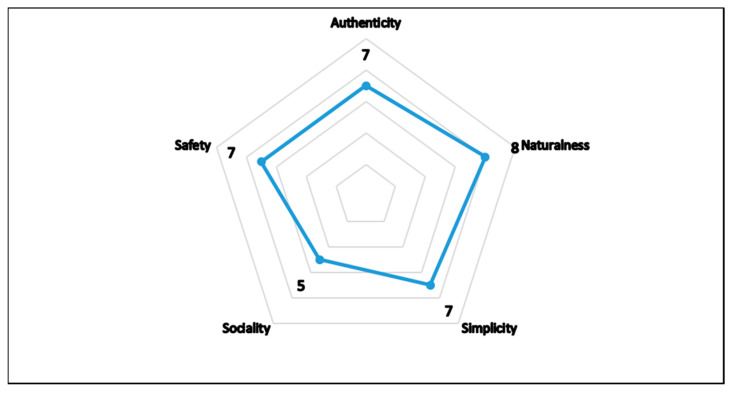
Assets of the trends related to CPL cheese (scale: 1 = absent; 10 = fully satisfying).

**Table 1 foods-12-04339-t001:** Physical and sensorial features of Caciocavallo Podolico Lucano cheese.

Parameter	Description
Shape	Pear shape with a slightly elongated head
Weight	Weight ranging from 2 to 3 kg
Diameter	The largest diameter of this cheese is between 20 and 22 cm
Height	Height, including the head, between 25 and 30 cm
Surface	Characterised by a thin and smooth crust in less seasoned products, with cracks that increase with seasoning
Colour	Straw yellow in less ripened cheese, darker yellow colour in more ripened cheese
Texture	Homogeneous and soft texture with rare holes in young products, firmer and grainy structure in more ripened ones
Taste	Sweet, oily, salty, slightly spicy taste in less seasoned cheese, various levels of spiciness in more ripened cheese
Smell	Fresh grass, berries, fruity, floral, wet straw, hay; seasoned cheese is very aromatic

**Table 2 foods-12-04339-t002:** Gross composition (g/100 g of cheese) of Caciocavallo Podolico Lucano cheese (averages ± SEMs).

Ripening Time	Moisture	Protein	Fat	Lactose	Ash	NaCl
6 months	33 ± 1	31 ± 1	30 ± 1	0	5.9 ± 0.3	2.3 ± 0.2
12 months	29 ± 1	32 ± 1	31.7 ± 1.5	0	6.6 ± 0.3	2.6 ± 0.2
Significance	ns	ns	ns		ns	ns

SEM = standard error of the mean; ns = not significant.

**Table 3 foods-12-04339-t003:** Fatty acid profile (g/100 g of cheese) of Caciocavallo Podolico Lucano cheese (averages ± SEMs).

Fatty Acids	Ripening Time		Significance
	6 Months	12 Months	
C4:0	1.02 ± 0.04	1.1 ± 0.1	ns
C6:0	0.64 ± 0.04	0.69 ± 0.05	ns
C7:0	0.004 ± 0.0001	0.005 ± 0.0001	ns
C8:0	0.32 ± 0.02	0.37 ± 0.03	ns
C10:0	0.66 ± 0.05	0.8 ± 0.1	ns
C11:0	0.08 ± 0.01	0.09 ± 0.01	ns
C12:0	0.7 ± 0.1	0.9 ± 0.1	ns
*iso* C13:0	0.016 ± 0.001	0.017 ± 0.002	ns
*anteiso* C13:0	0.021 ± 0.002	0.021 ± 0.002	ns
*iso* C14:0	0.07 ± 0.01	0.06 ± 0.01	ns
C14:0	2.7 ± 0.1	3.2 ± 0.2	ns
*iso* C15:0	0.018 ± 0.0005	0.019 ± 0.002	ns
*anteiso* C15:0	0.12 ± 0.02	0.13 ± 0.01	ns
C14:1c9	0.22 ± 0.02	0.21 ± 0.03	ns
C15:0	0.39 ± 0.05	0.45 ± 0.03	ns
*iso* C16:0	0.12 ± 0.01	0.12 ± 0.01	ns
C16:0	7.8 ± 0.4	8.6 ± 1	ns
*iso* C17:0	0.13 ± 0.01	0.15 ± 0.02	ns
C16:1 trans9	0.028 ± 0.003	0.04 ± 0.01	ns
C16:1 cis7	0.07 ± 0.01	0.068 ± 0.005	ns
*anteiso* C17:0	0.17 ± 0.02	0.19 ± 0.02	ns
C16:1 cis9	0.38 ± 0.03	0.34 ± 0.03	ns
C17:0	0.24 ± 0.03	0.27 ± 0.02	ns
*iso* C18:0	0.021 ± 0.003	0.021 ± 0.003	ns
C17:1 cis10	0.08 ± 0.01	0.067 ± 0.003	ns
C18:0	3.4 ± 0.1	3.8 ± 0.5	ns
C18:1 trans4	0.004 ± 0.001	0.004 ± 0.0005	ns
C18:1 trans5	0.004 ± 0.001	0.003 ± 0.001	ns
C18:1 trans6 + C18:1 trans8	0.065 ± 0.005	0.07 ± 0.01	ns
C18:1 trans9	0.072 ± 0.005	0.07 ± 0.01	ns
C18:1 trans10	0.08 ± 0.02	0.08 ± 0.01	ns
C18:1 trans11 (VA)	0.6 ± 0.1	0.9 ± 0.3	ns
C18:1 trans12	0.09 ± 0.01	0.08 ± 0.01	ns
C18:1 trans13 + C18:1 trans14	0.18 ± 0.01	0.21 ± 0.03	ns
C18:1 cis9	5.8 ± 0.2	5.2 ± 0.2	ns
C18:1 trans15 + C18:1 cis10	0.3 ± 0.1	0.16 ± 0.05	ns
C18:1 cis11	0.16 ± 0.01	0.14 ± 0.01	ns
C18:1 cis12	0.05 ± 0.13	0.04 ± 0.01	ns
C18:1 cis13	0.022 ± 0.001	0.020 ± 0.001	ns
C18:1 trans16 + C18:1 c14	0.101 ± 0.005	0.12 ± 0.02	ns
C18:2 trans9 trans12	0.014 ± 0.004	0.02 ± 0.01	ns
C18:2 cis9 trans13	0.050 ± 0.003	0.05 ± 0.01	ns
C18:2 cis9 trans12 + C18:2 trans8 cis12	0.021 ± 0.001	0.022 ± 0.003	ns
C18:1 cis16	0.023 ± 0.001	0.024 ± 0.003	ns
C18:2 trans9 cis12	0.009 ± 0.001	0.014 ± 0.003	ns
C18:2 trans11 cis15	0.09 ± 0.03	0.12 ± 0.04	ns
C18:2 cis9 cis12 (LA) n6	0.5 ± 0.1	0.4 ± 0.1	ns
C18:2 cis9 cis15 n3	0.006 ± 0.001	0.006 ± 0.001	ns
C20:0	0.0721 ± 0.01	0.08 ± 0.01	ns
C18:3 cis6 cis9 cis12 n6	0.008 ± 0.001	0.006 ± 0.001	ns
C20:1 cis9	0.003 ± 0.001	0.004 ± 0.001	ns
C20:1 cis11	0.016 ± 0.002	0.013 ± 0.002	ns
C18:3 cis9 cis12 cis15 n3	0.20 ± 0.03	0.28 ± 0.04	ns
C18:2 cis9 trans11 (CLA) (RU)	0.30 ± 0.04	0.4 ± 0.1	ns
C18:2 trans9 cis11 (CLA)	0.022 ± 0.003	0.029 ± 0.005	ns
C18:2 trans10 cis12 + C21:0 (CLA)	0.003 ± 0.001	0.003 ± 0.001	ns
C18:2 cis9 cis11 (CLA)	0.015 ± 0.004	0.02 ± 0.01	ns
C18:2 trans12 trans14 + C18:2 cis11 cis13 (CLA)	0.002 ± 0.001	0.006 ± 0.003	ns
C18:2 trans11 trans13 (CLA)	0.006 ± 0.001	0.009 ± 0.002	ns
C18:2 trans9 trans11 (CLA)	0.012 ± 0.001	0.008 ± 0.001	ns
C20:2 cis11 cis14 n6	0.007 ± 0.001	0.007 ± 0.001	ns
C20:3 cis5 cis8 cis11	0.013 ± 0.003	0.02 ± 0.01	ns
C22:0	0.032 ± 0.005	0.04 ± 0.01	ns
C20:4 cis5 cis8 cis11 cis14 n6	0.038 ± 0.003	0.032 ± 0.005	ns
C23:0	0.015 ± 0.002	0.020 ± 0.004	ns
C24:0	0.020 ± 0.003	0.025 ± 0.004	ns
C20:5 c5c8c11c14c17 n3 (EPA)	0.016 ± 0.003	0.023 ± 0.002	ns
C24:1c15	0.004 ± 0.001	0.005 ± 0.001	ns
C26:0	0.011 ± 0.002	0.014 ± 0.003	ns
C22:5 cis7 cis10 cis13 cis16 cis19 n3 (DPA)	0.031 ± 0.004	0.044 ± 0.004	ns
C22:6 cis4 cis7 cis10 cis13 cis16 cis19 n3 (DHA)	0.003 ± 0.001	0.005 ± 0.001	ns
BCFAs	0.68 ± 0.03	0.74 ± 0.04	ns
OCFAs	0.72 ± 0.03	0.82 ± 0.04	ns
ΣOBCFAs	1.4 ± 0.1	1.6 ± 0.1	ns
SCFAs (C4-C8)	1.98 ± 0.05	2.2 ± 0.1	ns
MCFAs (C10-C16)	13.5 ± 0.4	14.7 ± 0.5	ns
LCFAs (C17-C26)	13.1 ± 0.3	13.3 ± 0.4	ns
SFAs	19 ± 1	21 ± 1	ns
MUFAs	8.4 ± 0.2	7.8 ± 0.4	ns
PUFAs	1.40 ± 0.05	1.5 ± 0.2	ns
Σn6	0.6 ± 0.1	0.5 ± 0.1	ns
Σn3	0.25 ± 0.04	0.36 ± 0.05	ns
PUFA-n6/n3	2.8 ± 0.9	1.4 ± 0.3	ns

SEM = standard error of the mean; ns = not significant; VA = vaccenic acid; RU = rumenic acid; CLA = conjugated linoleic acid; EPA = eicosapentaenoic acid; DPA = docosapentaenoic acid; DHA = docosahexaenoic Acid; BCFAs = branched-chain fatty acids, the sum of iso and anteiso BCFA isomers (13:0–18:0); OCFAs = odd-chain fatty acids, sum of C11:0 to C23:0; ΣOBCFA = sum of BCFAs and OCFAs; SFAs: saturated fatty acids; MUFAs: monounsaturated fatty acids; PUFAs: polyunsaturated fatty acids; SCFAs: short-chain fatty acids; MCFAs: medium-chain fatty acids; LCFAs: long-chain fatty acids; Σn6 = sum of PUFA omega-6 fatty acids; Σn3 = sum of PUFA omega-3 fatty acids.

**Table 4 foods-12-04339-t004:** Fat-soluble vitamins and cholesterol (mg/kg of cheese) in Caciocavallo Podolico Lucano cheese (averages ± SEMs).

Ripening Time	Total Retinol	α-Tocopherol	Cholesterol
6 months	2.0 ± 0.1	9 ± 1	872 ^$^ ± 55
12 months	1.9 ± 0.2	7 ± 2	1089 ^&^ ± 83
Significance	ns	ns	0.057

SEM = standard error of the mean; ns = not significant; ^$,&^ symbols in columns indicate a tendency for *p* > 0.05.

**Table 5 foods-12-04339-t005:** Total polyphenol content (TPC) and total antioxidant capacity (CAT) of Caciocavallo Podolico Lucano cheese (averages ± SEMs).

Ripening Time	TPC (g GAE/kg of Cheese)	CAT	
		FRAP	TEAC
		(mmol FeSO_4_/kg of Cheese)	(mmol TROLOX/kg of Cheese)
6 months	4.1 ± 0.2	1.4 ± 0.1	67 ± 3
12 months	5 ± 1	1.4 ± 0.2	69 ± 9
Significance	ns	ns	ns

SEM = standard error of the mean; CAT = total antioxidant capacity; TPC = total phenolic content; GAE = gallic acid equivalent; FRAP = Ferric Reducing Antioxidant Power; TEAC = Trolox equivalent antioxidant capacity; ns = not significant.

**Table 6 foods-12-04339-t006:** Nutritional indexes of Caciocavallo Podolico Lucano cheese (averages ± SEMs).

Index		Ripening Time		Significance
		6 Months	12 Months	
HPI ^1^	Health-Promoting Index	0.51 ± 0.03	0.44 ± 0.05	ns
GHIC-7 ^2^	General Health Index of Cheese-7	32 ± 3	39 ± 4	ns

SEM = standard error of the mean; ns = not significant; ^1^ HPI = ΣUFA/[C12:0 + (4 × C14:0) + C16:0] (Chen, 2004 [45]); ^2^ GHIC-7= Σ score of C4:0 + C15:0 + C17:0 + C18:3c9c12c15 + C18:2c9t11 + CAT + TP.

## Data Availability

Data are contained within the article.

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
