# Peer review of "Caciocavallo Podolico Cheese, a Traditional Agri-Food Product of the Region of Basilicata, Italy: Comparison of the Cheese’s Nutritional, Health and Organoleptic Properties at 6 and 12 Months of Ripening, and Its Digital Communication"

_foods, 2023, doi:10.3390/foods12234339_

Round 1
Reviewer 1 Report
Comments and Suggestions for Authors
This study evaluated evaluate the effects of the ripening time (6 vs 12 months) on the CPL cheese characteristics to highlight the peculiar traits of this TAP cheese in terms of diverse indexes. This is an interesting study. However, some clarification and improvements should be made.
1. The author said that the CPL is “produced exclusively with raw milk from Podolian cows”. So, I'm curious what is the difference between Podolian milk and Holstein milk or any other milk? What are the advantages of using Podolian milk?
2. why the authors comparing the effects of the ripening time (6 vs 12 months) on the cheese characteristics? Why not select the fresh cheese or ripening 3 months?
3. I’m interested in the sensory fingerprint characterize detection. Results from sensory data fusion (Figure 6), combining color, smell and taste, like a human panel, objectively underline the clear difference between the two groups of CPL at two different ripening times. Can the authors attempt to explain or discuss the reasons for this difference?
4. It is recommended that the author explain the unique advantages or characteristics of CPL cheese in the introduction, such as nutritional value, taste, health value, etc., rather than just emphasizing its uniqueness.
5. The author used “$, & symbols” in table4 to indicate significant differences. This usage seems to be rare. It is common to use “*, **”.
6. It seems that most of the composition of CPL cheese is not significantly changed at 6 and 12 months of ripening. So, what factors affected the sensory fingerprint characterize? Authors are advised to provide more explanation and discussion.
Author Response
Responses to the reviewer 1
The authors thank the reviewer for the time he dedicated to us, the responses to the requests are reported below.
- The author said that the CPL is “produced exclusively with raw milk from Podolian cows”. So, I'm curious what is the difference between Podolian milk and Holstein milk or any other milk? What are the advantages of using Podolian milk?
Reply 1.
I thank the reviewer for this question.
The difference between Podolica and Holstein milk lies in their fat and protein content, which varies due to the breed's genotype, breeding and feeding systems. Podolica cows are raised using an extensive system and fed on pasture, while Holstein cows are raised using an intensive system and fed dry feed in the stable. These factors greatly affect the quality of the milk produced. Many studies have highlighted the differences in milk between breeds (refer to the sources below). One of the benefits of using Podolica cattle milk is that the breed can thrive and produce in marginal areas, making the best use of the plant resources available in their environment. Furthermore, the breed has an ecosystem service role (Braghieri et al., 2015).
No studies have been carried out comparing the Podolica breed with the Friesian but the effect of the breed has been the subject of studies.
Braghieri, A.; Pacelli, C.; Bragaglio, A.; Sabia, E.; Napolitano, F. The Hidden Costs of Livestock Environmental Sustainability: The Case of Podolian Cattle. In The Sustainability of Agro-Food and Natural Resource Systems in the Mediterranean Basin; Vastola, A., Ed.; Springer International Publishing: Cham, 2015; pp. 47–56 ISBN 978-3-319-16356-7.
De Marchi, M., Bittante, G., Dal Zotto, R., Dalvit, C., & Cassandro, M. (2008). Effect of Holstein Friesian and Brown Swiss breeds on quality of milk and cheese. Journal of Dairy Science, 91(10), 4092-4102.
Yang, T. X., Li, H., Wang, F., Liu, X. L., & Li, Q. Y. (2013). Effect of cattle breeds on milk composition and technological characteristics in China. Asian-Australasian Journal of Animal Sciences, 26(6), 896.
Kebede, E. (2018). Effect of cattle breed on milk composition in the same management conditions. Ethiopian Journal of Agricultural Sciences, 28(2), 53-64.
- why the authors comparing the effects of the ripening time (6 vs 12 months) on the cheese characteristics? Why not select the fresh cheese or ripening 3 months?
Reply 2. The authors conducted a study comparing 6-month-old and 12-month-old cheeses because these are the types that are usually requested by consumers, as stated in line 124. This information was obtained from the sales records of the cheese shop and confirmed through another project on the survey that evaluated the perception of ecosystem services in the Podolian farming system, located in the southern marginal areas of Italy. Additionally, it takes a long time to mature raw milk cheeses to ensure their safety for consumption, as stated by the E.C. in 2004.
References: European Commission (2004). Regulation (EC) No. 853/2004 of the European Parliament and of the Council of 29 April 2004 establishing specific hygiene rules for foods of animal origin. Gazz Uff, 55-205.
- 3. I’m interested in the sensory fingerprint characterize detection. Results from sensory data fusion (Figure 6), combining color, smell and taste, like a human panel, objectively underline the clear difference between the two groups of CPL at two different ripening times. Can the authors attempt to explain or discuss the reasons for this difference?
Reply 3. Several biochemical events occur during ripening that alters sensory properties of the manufactured cheese, which include three main reactions: metabolism of the residual lactose, lactate and citrate, proteolysis and lipolysis (McSweeney, 2011). Enzymes that participate in the ripening process come from various sources. From the milk, lipoprotein lipase survives pasteurisation to participate in lipolysis. From coagulation, some of the rennet enzyme “chymosin” is retained to participate in proteolysis. The starter bacteria, in addition to their primary function of fermentation, provide proteinase and esterase enzymes, among other enzymes. NSLAB and secondary cultures are among the major contributors to the ripening process with their strong sensory impact (Clark, Costello, Drake, & Bodyfelt, 2009). Furthermore, no less important, the penetration of salt also significantly affects the organoleptic properties, first of all, the taste. In our work, many chemical characteristics, such as the amino acid profile and the volatile components, were not investigated, but were "captured" by artificial senses through the evaluation of the colorimetric profile, the volatile profile and the dissolved compounds (colour, odour and taste).
P.L.H. McSweeney Cheese: Biochemistry of cheese ripening Encyclopedia of dairy sciences (2nd ed.) (2011), pp. 667-674 https://doi.org/10.1016/B978-0-12-374407-4.00080-7
- Clark, M. Costello, M.A. Drake, F. Bodyfelt The sensory evaluation of dairy products The Sensory Evaluation of Dairy Products (2009) https://doi.org/10.1007/978-0-387-77408-4
- 4. It is recommended that the author explain the unique advantages or characteristics of CPL cheese in the introduction, such as nutritional value, taste, health value, etc., rather than just emphasizing its uniqueness.
Reply 4. The aim of this work is to highlight the characteristics of the cheese, including its nutritional value, taste, and health benefits. However, due to the limited literature available on the topic, only known characteristics could be explained in the introduction.
- The author used “$, & symbols” in table4 to indicate significant differences. This usage seems to be rare. It is common to use “*, **”.
Reply 5. The symbols “$, &" used in Table 4 indicate a trend towards significance p= 0.10, this is clarified in the note below Table 4. For greater clarity, the note below the table becomes: $, & symbols in columns indicate a trend of the differences for p = 0.10.
- It seems that most of the composition of CPL cheese is not significantly changed at 6 and 12 months of ripening. So, what factors affected the sensory fingerprint characterize? Authors are advised to provide more explanation and discussion.
Reply 6 For the point n.3 and point n.6 we have a unique answer.
Reviewer 2 Report
Comments and Suggestions for Authors
The authors have done great job, really appreciable. I strongly recommend the work.
Author Response
Reviewer 2
The authors have done great job, really appreciable. I strongly recommend the work.
Reply: The authors thank the reviewer for the time he dedicated to us.
Reviewer 3 Report
Comments and Suggestions for Authors
I think this paper needs a substantial revision. To help authors I merged my review together with the submitted paper with annotations.

Author Response
Reviewer 3
Courteous reviewer,
We thank you for the time you dedicated to improving our work. The various co-authors answered the questions by trying to explain, first of all, the objective of the work, which was to raise awareness of this little-studied product by addressing the problem from various aspects.
For the first time, a socio-semiotic evaluation was combined with chemical evaluations. However, in this era in which communication through new mass media plays a role, we have tried to understand the effective strategies to promote a Traditional Agri-food Product with these tools.
The various answers are reported in correspondence with the indications (lines) provided by the auditor.
Abstract
Line 30 and 34:
I understand what the authors mean by firgerprint, but unless it is previously
explained, it does not make sense (at least in the abstract). Suggest reformulation.
Reply 30 and 34: We understood the meaning of what you wanted to say and we preferred to replace the term ”sensory fingerprint” with “artificial sensory profile” or “organoleptic fingerprint
Line 31:
is web communication a chemical parameter? As it is written in the paragraph context, it
looks like that. I suggest some reformulation
Reply 31: The reviewer's advice is accepted, and the sentences have been reformulated as follows.
Additionally, this study represents the first analysis of cheese-related digital communication and trends online.
Line 33: it is lactose free? I agree in part, because 100% is impossible! Shoud refer “at the detection
level”.
Reply 33: Everything has been clarified as reported in the text of paragraph 3.1 (line 322) : “In Table 2, the lactose content was declared equal to 0g/100g of cheese, based on the guidance document on the rounding rules applicable to the declaration of nutrients (European Commission Health and Consumers Directorate-General, December 2012: Guidance document for competent authorities for the control of compliance with EU legislation on: Regulation (EU) No 1169/2011, Council Directive 90/496/EEC and Directive 2002/46/EC). According to this guidance document, if the sugar content in a food is less than or equal to 0.5g/100g, it can be declared as 0g/100g in the nutrition table. Given the importance of this declaration, also in light of the fact that lactose intolerance is very widespread and since the official method used for the determination of lactose [30] did not report any limits of detection and quantification, the samples of CPL were also analysed with another validated analytical method (Idda et al, 2018). This method which involves the determination of lactose by gas chromatography with a Limit of Detection (LOD) of 0.54mg/kg of cheese and a Limit of Quantification (LOQ) of 1.68mg/kg of cheese, confirmed that the lactose content in CPL is on average 0.001g/100g, therefore lower than the limit set by the aforementioned ministerial note, so that the cheese can be declared “naturally lactose-free”.”
The reference cited was developed by a co-author of this work, and was reported in the references
- Idda I., Spano N., Addis M., Galistu G., Ibba I., Nurchi V.M., Pilo M.I., Scintu M.F., Piredda G., Sanna G. 2018. Optimization of a newly established gas-chromatographic method for determining lactose and galactose traces: Application to Pecorino Romano cheese. Journal of Food Composition and Analysis, 74, 89-94).
Introduction
Line 54: Names of institutions should refer Italy, e.g., the Ministry of Agriculture of Food Sovereignty
and Forests (MASAF). Is it the italian one?
Reply 54: the name of the ministry was reported in Italian
Lines 55- 57: I understand the meaning, but language correction is needed.
Reply 55- 57: The sentence has been improved.
Recently data from the XX ISMEA–Qualivita report [4] shows increasing demand for local or traditional foods, as they are often perceived to be of higher quality [5], more sustainable [3], and with a solid cultural identity [6], than industrial foods
Line 78: “and from the previous”,… rewriting is suggested
Reply 78 now 85-869: the sentence has been rewritten
There are very few studies focused on TAPs and their characterisation; thus, the need to expand knowledge of TAP cheeses emerges.
Line 83: “Slow Food praesidium”,…
a reference to this foundation would be helpful for readers
Reply 83: The link to the foundation is added in the text:
https://www.fondazioneslowfood.com/it/
Line 107 “ pedo climatic conditions”…must have a hyphen?
Reply 107: No. The hyphen has been removed.
Line 92: “sensory properties of cheese”…again, the cheese has no sensory properties. Please correct
here and elsewhere
Reply 92: Thank you for your time and for your suggestions, which certainly improve the article. Sensory has been replaced with organoleptic in the title and throughout the text.
Line 100: “∑PUFA”… please first write in full, then the abbreviation. Also, do you mean total PUFA?
Please correct.
Reply 100: Yes. It was reported in the text total PUFA.
Line 110: what do you mean by “sensorial fingerprint”? If fingerprint is something that makes a
product unique, can sensory analysis provide such a desideratum?
Reply 110: Instrumental sensory analysis provides a specific, unique and non-repeatable fingerprint of the sample.
Materials and methods
Section 2.1. Please, clearly specify number of producers, and number of cheeses per producer and per
ripening time, as well as number of analytical replicates per cheese, providing the total amount of
cheeses analysed.
Reply 2.1. The information requested by the reviewer has been reported:
1) During the sampling process, we selected all 5 CPL producers who have all the required documentation to deliver their products to a specialised shop for sale. Unfortunately, we couldn't sample any "not completely regular for sale" production since it wasn't part of our scope. In fact, on the same day, CPL with 6 months and 12 months ripened belonging to all the previously identified manufacturing producers were purchased at the cheese shops.
The authors would like to point out that this work, in addition to characterising the CPL on several aspects and evaluating differences between maturation times, has the long-term objective, not the subject of this work, of raising awareness, bringing out and regularising as many producers as possible with the help from regional authorities and also with the right communication of the product.
2) six cheeses per producer: three at 6 months and three at 12 months of ripening
3) two analytical replicates for the same cheese
4) the total amount of cheeses was 30
To our regret, it was impossible to increase the number of samples per ripening due to the high cost of the product (from 26-45 euros/kg) depending on the ripening.
In MM was added: For each of the 5 producers, 3 samplings were carried out for each ripening period.
Line 122: replace hyphen (--) by minus (‒) sign
Reply 122: done
Lines 116- 118: The study involved cheeses with a maturation period of 6 and 12 months.
Consequently, the diet of the animals, raised on natural pastures, was very different. Therefore, it is
not only the curing time that is being studied, but also the variation in the raw material. How was this
taken into account?
Reply 116-118: The experiment aimed to evaluate the characteristics of CPL cheese at different ripening periods, purchased on the same day. To achieve this, the cheeses were sampled on the same day to evaluate what the consumer would have available at the time of purchase. Cheese aged for 6 and 12 months, respectively, was produced using milk from cows grazing in autumn (November) and spring (May). From observations made by the zootechnical group, at our latitudes, the pastures of November and May are not very different. The temperatures are not extreme; in fact, they allow the use of green grass in quantities adequate to cover the maintenance and production requirements of the animals in order to guarantee the production and quality levels of the milk. Natural pastures are very different when the grass is dry/ absent and green. From what we have observed in Podolica farms, we believe that the effect of the period can be considered negligible.
Line 152: “according to [34] method”... should be “according to Jiang et al. [34] method.”. The same
applies to all similar situations.
Reply 152: done
Line 255: “6 and 12 month aged cheese”… why the hyphens in this part of the text?
Reply 110: done
Lines 260- 262: a language correction is necessary.
Reply 260- 262: done
Lines 270-271: if there is only one factor (curing time) with two levels (6 and 12 months), then
ANOVA is not the most appropriate procedure and student's t tests should be applied.
Reply 270-271: Thank you for the clarification provided by the reviewer. We have conducted an analysis on some of the variables measured using the recommended test. As expected, the means and p-values remain unchanged. Therefore, we have decided to retain the statistical procedure used since the outcome remains consistent.
We concur with the reviewer's observation that the data variability could be influenced by the type of non-industrial product being analysed and the number of available samples.
Line 275: usually PCA is used instead of P.C.A.
Reply 275: done
Results and discussion
Line 285: what is "good" content? Needs rewriting.
Reply 285: the sentences have been reformulated.
.
Lines 287- 294. If gross composition differed not significantly, then there was no difference. If
increases and decreases are tested and p values are greater than 0.11, than there are no changes.
Reply 287-294: I confirm that there are no significant differences in the tables, as can be seen from the p-Value. Taking the referee's observation into account, we do not use "increase" and "decrease" in the results and discussion section.
Table 2: as far as I see, there are no significant differences and the tests should be student's t tests, in
accordance with my previous comments.
Reply Table 2: Table 2 reports the mean, esm and the p_value of the procedure used. The lack of significance is evident, as already reported in the text (see also Reply 270-271). The lack of differences was made clear in the discussions.
Line 300: about lactose
The authors are discussing a result, or is a statement that needs a reference?
Reply 300, now 311: the word “stewing” has been replaced with the word “acidification”
References have been added in:
“Generally, lactose is not completely hydrolysed during the curd's acidification phase. The amount remaining in fresh cheese depends on several factors, the most important being the type of starter used in the cheese-making process and the duration of the curd acidification phase (Portnoi et al., 2009; Idda et al. 2018; 24). The residual lactose content is further reduced during cheese ripening; long-ripened cheeses are generally naturally lactose-free [Gille et al. 2018].”
Reference
Portnoi, P. A., & MacDonald, A. (2009). Determination of the lactose and galactose content of cheese for use in the galactosaemia diet. Journal of human nutrition and dietetics, 22(5), 400-408.
Idda I., Spano N., Addis M., Galistu G., Ibba I., Nurchi V.M., Pilo M.I., Scintu M.F., Piredda G., Sanna G. 2018. Optimization of a newly established gas-chromatographic method for determining lactose and galactose traces: Application to Pecorino Romano cheese. Journal of Food Composition and Analysis, 74, 89-94.
[24] Busetta, G., Garofalo, G., Barbera, M., Di Trana, A., Claps, S., Lovallo, C., ... & Settanni, L. (2023). Metagenomic, microbiological, chemical and sensory profiling of Caciocavallo Podolico Lucano cheese. Food Research International, 169, 112926.
Gille, D., Walther, B., Badertscher, R., Bosshart, A., Brügger, C., Brühlhart, M., ... & Egger, L. (2018). Detection of lactose in products with low lactose content. International Dairy Journal, 83, 17-19.
Line 305: instead of “a significant outcome, namely that CPL is a “lactose
free cheese”, a suggest writing “a significant outcome, namely that CPL is a “lactose free” cheese” considering the detection level of the analytical procedure.”.
Reply 305: Based on the level of detection of lactose (see Reply 33), it can be concluded that CPL is naturally lactose-free.
Lines 307- 314: doesn't this reasoning about lactose apply to many matured cheeses?
Reply 307-314: As reported in lines 318-321, the factors that influence the presence of lactose in cheeses are various, including the starter culture used during the cheese-making process and the maturation time. So, we cannot say this reasoning applies to all mature cheeses (see example).
For example: In Fiore Sardo, unlike Pecorino Romano, which is lactose-free at 24 hours (Idda et al. 2018), lactose residues remain even after 3.5 months of maturation. The difference lies in the bacterial species of the starter (thermophilic lactobacilli in the case of Pecorino Romano and mesophilic lactococci in the case of Fiore Sardo).
Lines 318 and 322: “Perna et al. [23] and Pizzillo et al. [17]” I agree with this way of referring other works, which should be extended to the complete text as referred above.
Reply 318 and 322: I fully agree with the referee and confirm that all the scientific papers available on CPL have been found and analysed; in particular, only the studies in which the cheesemaking process was the traditional one were used. If, during the discussion, no comparisons with other works are reported, it is because no scientific works on the topic are available in the literature. This situation was one of the reasons that pushed us to carry out this work. A very recent reference has been added.
Recently, Natrella et al. (Natrella et al., 2023) did not detect significant differences in gross composition between Caciocavallo cheeses aged 180 and 340 days. The moisture, fat and protein values of CPL are in agreement with what was reported in the aforementioned study.
The bibliographic reference has been added in the references section
Natrella, G., De Palo, P., Maggiolino, A., & Faccia, M. (2023). A Study on Milk and Caciocavallo Cheese from Podolica Breed in Basilicata, Italy. Dairy, 4(3), 482-496.
Lines 333-334: so, here, in comparison with gross composition, the authors say that there are no
differences, in face of the same statistical p>0.11 support! Facing similar p values must lead to the
same conclusion of not rejecting the null hypothesis (see my comments above in relation to gross
comp osition). Therefore, until line 412 the text needs to be restructured.
Reply 333-334: Taking into account the referee's observation, the results and discussions have been reformulated.
Table 3:
as far as I can see, there are no differences between ripening times. And if this is true, why
are the authors referring to different values in the text?
Reply Table 3. Taking into account the referee's observation, the results and discussions have been reformulated. The words increase and decrease are deleted in discussion of our results.
Lines 416- 420 and Table 4. But these are mainly non significant differences! As far as I see,
significant differences are observed only for cholesterol. Needs to be restructured
Reply 416- 420 and Table 4. Taking into account the referee's observation, the results and discussions have been reformulated.
Table 5: CAT is referred, but the table does not show CAT data.
Reply Table 5: the CAT reference line has been added to the table
Table 6: Again no significant differences were observed.
Reply Table 6: Taking into account the referee's observation, the results and discussions have been reformulated.
Lines 556-557: Aren't those numerous factors also affecting the other aspects already discussed?
Reply 556-557: Several biochemical events occur during ripening that alter sensory properties of the manufactured cheese, which include three main reactions: metabolism of the residual lactose, lactate and citrate, proteolysis and lipolysis (McSweeney, 2011). Enzymes that participate in the ripening process come from various sources. From the milk, lipoprotein lipase survives pasteurisation to participate in lipolysis. From coagulation, some of the rennet enzyme “chymosin” is retained to participate in proteolysis. The starter bacteria, in addition to their primary function of fermentation, provide proteinase and esterase enzymes, among other enzymes. NSLAB and secondary cultures are among the major contributors to the ripening process with their strong sensory impact (Clark, Costello, Drake, & Bodyfelt, 2009). Furthermore, no less important, the penetration of salt also significantly affects the organoleptic properties, first of all, the taste. In our work, many chemical characteristics, such as the amino acid profile and the volatile components, were not investigated, but were "captured" by artificial senses through the evaluation of the colorimetric profile, the volatile profile and the dissolved compounds (colour, odour and taste).
P.L.H. McSweeney Cheese: Biochemistry of cheese ripening Encyclopedia of dairy sciences (2nd ed.) (2011), pp. 667-674 https://doi.org/10.1016/B978-0-12-374407-4.00080-7
- Clark, M. Costello, M.A. Drake, F. Bodyfelt The sensory evaluation of dairy products The Sensory Evaluation of Dairy Products (2009) https://doi.org/10.1007/978-0-387-77408-4
Request: ABOUT SENSORY FINGERPRINTS
In section 3.5 and figures 3-6, many samples are seen. What are these samples? Replications? Cheeses from different producers?
Reply: In section 3.6, figures 3-6 show the samples collected as described in section 2.1. (Survey design and cheese sample collection). Five producers, three replicates for cheese sample for each period of ripening (6 and 12 months). Each sample for the e-senses analysis was analysed 16 for e-eye, 30 for e-tongue, and 4 for e-nose. Then, a small number of analytical replications were chosen for each data set (depending on the stability of the sensors). For data fusion, 3 replicates for each sample were used.
Lines 574- 575: What is sensory data fusion? How is it defined and obtained?
Reply 574-575: In general, fusion of data from complementary sensors, responding to different signature phenomena, may increase the probability of correct classification; therefore, characteristics that are not classified by one sensor may be apparent or measured by another [Di Rosa et al., 2017; Banerjee et al., 2016]. Data fusion processes are often categorised in a three-level model, distinguishing low, intermediate, and high level fusion [Di Rosa et al., 2017; Elmenreich, 2002]. In this study, intermediate fusion level was investigated. Intermediate level fusion, first extracts some relevant features from each data source separately, and then concatenates them into a single array that is used for multivariate classification and regression [27, 34]. The datasets already used were scaled, due to the different data magnitude, and simple concatenated in a global matrix with a total of 9996 variables (low-level fusion). To avoid redundant information and reducing data dimensionality, a feature extraction was carried out (mid-level fusion). The instrumental software allows to select automatically the best inputs to achieve a target task. With the novel dataset, a PC analysis was applied.
Di Rosa AR, Leone F, Cheli F, Chiofalo V (2017) Fusion of electronic nose, electronic tongue and computer vision for animal source food authentication and quality assessment—a review. J Food Eng 210:62–75. doi:10.1016/j.jfoodeng.2017.04.024
Banerjee R, Tudu B, Bandyopadhyay R, Bhattacharyya N (2016) A review on combined odour and taste sensor systems. J Food Eng 190:10–21. doi:10.1016/j.jfoodeng.2016.06.001
Elmenreich W (2002). Principles of sensor fusion. Sensor fusion in time-triggered systems, pp 7–16. https://mobile.aau. at/~welmenre/papers/elmenreich_Dissertation_sensorFusionIn-TimeTriggeredSystems.pdf
Request: Ellipses, were they calculated, or hand drawn?
Reply: Ellipses serve only to limit and identify the area in which the samples of a group are located and are drawn by hand.
Request: As far as I can see, exception made for e-tongue data, differences between groups are not as evident as the authors say. These differences need some statistical support, using some kind of discriminant technique, such as canonical variates analysis or MANOVA.
Reply: Data evaluation was expressed based on the discrimination index (DI), which gives an evaluation of the discrimination quality on the selected plan from the surface (non-Euclidean) between groups and the size of each group.
1- When groups are distincts, the discrimination index is calculated according to the following formula:
Di = 100 *[1 – [(Surface(A)+Surface(B)+Surface(C)) /(Total Surface) ]]
2- When groups overlap each other, the discrimination index is calculated according to the following
formula:
Di= -(Ʃ Intersection Surface/Total surface) * 100
For better understand the difference between groups, the Discrimination Index (DI) was added in the caption of each figure. In addition, section 2.10 “Statistical analysis” has been expanded and improved.
Throughout the text, all figures must be separated from units, e.g., 102 ºC, not 102ºC. , seconds,
minutes, etc.
Reply: done
The complete sections 3.5 and 3.6 Have the same problems as discussed for previous sections.
Reply: Taking into account the referee's observation, the results and discussions have been reformulated.

Reviewer 4 Report
Comments and Suggestions for Authors
The manuscript entitled „Caciocavallo Podolico cheese, a Traditional Agri-food Product (TAP) of Basilicata region: nutritional, health, sensory properties and digital communication” describes chosen properties of Caciocavallo Podolico Lucano cheese. Authors investigated among others the effects of ripening time on the quality of these cheese, including fatty acid profile, fat-soluble vitamins, cholesterol, total polyphenol content, total antioxidant capacity, nutritional indexes etc. The reviewer below provides comments and suggestions, minor as well major:
1. Title: the fullstop at the end of the title is unnecessary, there should be no abbreviation in the title (it is enough that it is explained in the text).
I would consider changing the title. The current one sounds like the title of a review article. I do not want to push, but the title may be misleading. If the authors think about the goal of the research and define it well, they may be able to change the title, e.g. to compare selected properties of... cheese after 6 and 12 months of storage.
2. In the abstract, please clearly indicate 2-3 conclusions from the research conducted.
3. Keywords need to be simplified and rephrased. These words are intended to be specific words that make it easier to find an article on a topic that interests the reader. Currently, the keywords are too general and extensive, giving little chance of finding this manuscript after its eventually publication.
4. L44, when authors write “in recent years….” a new reference should be put here, I mean, something from last 2-4 years, not from 2015 like reference [3] is.
5. In the Introduction, the authors write very interestingly about TAP in general. This content is interesting and important, but I feel that it is unnecessary in this article. The article focuses strictly on Caciocavallo Podolico cheese and I would suggest to focus on the information about this product and reduce a little the general information about Italian TAPs.
6. I suggest switch the order of graphics 1 and table 1. First, let the reader see what the cheese looks like, and then let the reader read the characteristics in the table. I also suggest to put this two graphics in the introduction section, not after the materials and methods.
7. L 94, if you start a new paragraph, do not refer directly to the text that is related to the previous paragraph. If you want to leave "This traditional cheese..." do not start new paragraph.
8. Figure 2 is wonderfully prepared (The photos in this diagram could be slightly bigger.), please just add references to the preparation steps if possible.
9. In my opinion "digital communication" and "trend on the web" parts should be explain and described. I would like to kindly ask the authors to expand on this issue in the introduction.
10. L112, I'm not sure if the term "survey" is appropriate here.
11. L149, Analytical research should be performed at least three times, not twice.
12. table 2 has not been announced.
13. 2.4. i don't feel that "retinol" term good corresponds with cheese...
14. Table 2, The authors explain that cheeses ripened for 12 months has greater moisture loss, which is why their NaCl, ash and fat content increased. Please explain why the protein content has not changed.
15. I like the explanation in lines 300-314, it has be pointed that this type of cheese are lactose free without any specific stages of technology, well done authors.
17. Table 6, The HPI for 6-month cheese is higher than for 12-month cheese, while in case of GHIC-7 is opposite, how can the authors explain this?
18. Chapter 3.7. is definitely not discussed enough. For me this information is not well known, this, let's call it "internet-virtual" topic of cheese is not close to me. First of all, authors should discuss the obtained results, discuss them with other authors (this has already been done) and explain them.
19. An extensive interpretation of the results should definitely be added here so that readers (including technologists who know cheese technology but are not familiar with the "web" field) can clearly understand what is going on and why is important.
20. In my opinion, the conclusion section should be written from the beginning. In the Conclusion section, please provide specific details of the differences and similarities in cheeses ripened for 6 and 12 months. Which cheese is more suitable for which consumer group and why. Please refer directly to the purpose of the work, has the goal been achieved?
21. Please clearly indicate (in the abstract and conclusion) what is novel aspect and what new knowledge this research brings to science and technology. Please describe in a strong and blunt manner the validity of the research presented here.
Comments on the Quality of English LanguageIt is always good to revise the manuscript by native speaker.
Author Response
Referee 4
The authors thank the reviewer for the time he dedicated to us, the responses to the requests are reported below.
- Title: the fullstop at the end of the title is unnecessary, there should be no abbreviation in the title (it is enough that it is explained in the text). I would consider changing the title. The current one sounds like the title of a review article. I do not want to push, but the title may be misleading. If the authors think about the goal of the research and define it well, they may be able to change the title, e.g. to compare selected properties of... cheese after 6 and 12 months of storage
Reply 1.
- The full stop at the end of the title has been removed.
- The acronym in the title has been eliminated.
- The auditor's advice was accepted. The title has been changed as follows. Caciocavallo Podolico cheese, a Traditional Agri-food Product of Basilicata region: comparison of the cheese’s nutritional, health and organoleptic properties at 6 and 12 months of ripening, and its digital communication
- In the abstract, please clearly indicate 2-3 conclusions from the research conducted.
Replay 2: The abstract and conclusions were rewritten according to the reviewer's advice
- Keywords need to be simplified and rephrased. These words are intended to be specific words that make it easier to find an article on a topic that interests the reader. Currently, the keywords are too general and extensive, giving little chance of finding this manuscript after its eventually publication
Reply 3. The search for a scientific paper is usually carried out on the words of the title and the keywords, also, taking into consideration the advice of the reviewer, the keywords have been reformulated as follows.
Keywords: typicality, TAP stretched cheese, cheese fatty acid, retinol, α-tocopherol, cholesterol, antioxidant capacity, nutritional indexes, organoleptic fingerprint, communication and trend on the web
- L44, when authors write “in recent years….” a new reference should be put here, I mean, something from last 2-4 years, not from 2015 like reference [3] is
Reply 4. The referee's advice is accepted, and the following words have been added "For more than five years to date"
- In the Introduction, the authors write very interestingly about TAP in general. This content is interesting and important, but I feel that it is unnecessary in this article. The article focuses strictly on Caciocavallo Podolico cheese and I would suggest to focus on the information about this product and reduce a little the general information about Italian TAPs.
Reply 5. The reviewer's advice was accepted, and some sentences were deleted (line 65-69) and others shortened (line 78-81).
- I suggest switch the order of graphics 1 and table 1. First, let the reader see what the cheese looks like, and then let the reader read the characteristics in the table. I also suggest to put this two graphics in the introduction section, not after the materials and methods.
Reply 6. The reviewer’s suggestion was accepted, and the recommended changes were made.
- L 94, if you start a new paragraph, do not refer directly to the text that is related to the previous paragraph. If you want to leave "This traditional cheese..." do not start new paragraph.
Reply 7. It has been corrected; a new paragraph is not started (see line 102).
- Figure 2 is wonderfully prepared (The photos in this diagram could be slightly bigger.), please just add references to the preparation steps if possible.
Reply 8. Placing the entire flowchart on a single page (Figure 2) did not allow the use of larger photos. References to the preparation steps are indicated in parentheses in the corresponding boxes of the flowchart.
- In my opinion "digital communication" and "trend on the web" parts should be explain and described. I would like to kindly ask the authors to expand on this issue in the introduction.
Reply 9. The reviewer's feedback was taken into account and as a result, paragraph 2.9 underwent a complete rewrite.
- L112, I'm not sure if the term "survey" is appropriate here.
Reply 10. The reviewer's suggestion is accepted, and the term “survey” is deleted
- L149, Analytical research should be performed at least three times, not twice.
Reply 11. We were able to perform twice. However, in the foods journal, we read many works that report analytical research performed twice.
- table 2 has not been announced.
Reply 12. Table 2 has been announced in line 305.
- 2.4. i don't feel that "retinol" term good corresponds with cheese...
Reply 13: Retinol is a fat-soluble vitamin found in cheese. There are various works that highlight this.
- Panfili, G., Manzi, P., & Pizzoferrato, L. (1994). High-performance liquid chromatographic method for the simultaneous determination of tocopherols, carotenes, and retinol and its geometric isomers in Italian cheeses. Analyst, 119(6), 1161-1165.
- Beliveau, A. R. (2012). Variations in carotenoids and retinol in milk and cheese from Jersey cows at an organic dairy compared to a conventional dairy over a pasture season. University of New Hampshire.
- Niro, S., Fratianni, A., Tremonte, P., Lombardi, S. J., Sorrentino, E., Manzi, P., & Panfili, G. (2022). Cis-trans retinol isomerisation: Influence of microorganisms during the production of pasta filata cheeses. International Dairy Journal, 133, 105441.
- Table 2, The authors explain that cheeses ripened for 12 months has greater moisture loss, which is why their NaCl, ash and fat content increased. Please explain why the protein content has not changed.
Reply 14. In paragraph 3.1, it is explained that the differences in the gross composition between the CPL at 6 and 12 months are not significant (p>0.10); however, a slight increase is observed for fat, ash and NaCl. Protein shows a very small increase at 12 months. The protein levels observed could be linked to the phenomenon of proteolysis and the lowering of its activity following the reduction of humidity during maturation. In their study on Ragusano cheese, a pasta filata cheese made from raw milk, Fallico et al. (2004) found a negative correlation between moisture and protein and a positive correlation between cheese moisture and proteolysis.
Reference has been added in the text and in the references.
Fallico, V., McSweeney, P. L. H., Siebert, K. J., Horne, J., Carpino, S., & Licitra, G. (2004). Chemometric analysis of proteolysis during ripening of Ragusano cheese. Journal of Dairy Science, 87(10), 3138-3152.
- I like the explanation in lines 300-314, it has be pointed that this type of cheese are lactose free without any specific stages of technology, well done authors.
Reply 15. In Table 2, the lactose content was declared equal to 0g/100g of cheese, based on the guidance document on the rounding rules applicable to the declaration of nutrients (European Commission Health and Consumers Directorate-General, December 2012: Guidance document for competent authorities for the control of compliance with EU legislation on Regulation (EU) No 1169/2011, Council Directive 90/496/EEC and Directive 2002/46/EC). According to this guidance document, if the sugar content in a food is less than or equal to 0.5g/100g, it can be declared as 0g/100g in the nutrition table. Given the importance of this declaration, also in light of the fact that lactose intolerance is very widespread and since the official method used for the determination of lactose (30) did not report any limits of detection and quantification, the samples of Caciocavallo Podolico were also analysed with another validated analytical method (Idda et al. 2018). This method which involves the determination of lactose by gas chromatography with a Limit of Detection (LOD) of 0.54mg/kg of cheese and a Limit of Quantification (LOQ) of 1.68mg/kg of cheese, confirmed that the lactose content in Caciocavallo Podolico is on average 0.001g/100g, therefore lower than the limit set by the aforementioned ministerial note, so that the cheese can be declared “naturally lactose-free”.
The reference cited was developed by a co-author of this work and was reported in the references
Idda I., Spano N., Addis M., Galistu G., Ibba I., Nurchi V.M., Pilo M.I., Scintu M.F., Piredda G., Sanna G. 2018. Optimization of a newly established gas-chromatographic method for determining lactose and galactose traces: Application to Pecorino Romano cheese. Journal of Food Composition and Analysis, 74, 89-94).
- 17. Table 6, The HPI for 6-month cheese is higher than for 12-month cheese, while in case of GHIC-7 is opposite, how can the authors explain this?
Reply 17. The opposite trend of the two indices lies in the different components considered for their calculation, as indicated in the caption of Table 6. The HPI value in the 12-month CLP can be explained by the primary biochemical changes that occur during the cheese ripening process, such as lipolysis. (McSweeney et al., 2004). This leads to an increase in AG in mature cheeses, causing the denominator of the HPI index to increase and the index itself to decrease.
In paragraph 3.5, the following was added: The HPI value in the 12-month CLP can be explained by the primary biochemical changes that occur during the cheese ripening process, such as lipolysis (McSweeney et al., 2004). This leads to an increase in AG in mature cheeses, causing the denominator of the HPI index to increase and the index itself to decrease.
Reference has been added in the text and in the references.
McSweeney, P. L. (2004). Biochemistry of cheese ripening. International journal of dairy technology, 57(2‐3), 127-144.
- Chapter 3.7. is definitely not discussed enough. For me this information is not well known, this, let's call it "internet-virtual" topic of cheese is not close to me. First of all, authors should discuss the obtained results, discuss them with other authors (this has already been done) and explain them.
Reply 18: The feedback of the reviewer was taken into account, and we have rewritten the material and methods (in paragraph 2.9) and rewritten and expanded the discussion (in yellow).
- An extensive interpretation of the results should definitely be added here so that readers (including technologists who know cheese technology but are not familiar with the "web" field) can clearly understand what is going on and why is important.
Reply 19. The reviewer's advice was accepted, and in paragraph 3.7, the results were specified, and the role of this type of analysis was clarified in the introduction, results and discussion (in yellow).
In the introduction was added: ]To effectively communicate about CPL cheese, it is important to evaluate how the product is currently being communicated through web channels. A useful approach to achieve this objective is to use socio-semiotic analysis (27). This method can help in identifying the best drivers for creating effective communication of this cheese
- 20. In my opinion, the conclusion section should be written from the beginning. In the Conclusion section, please provide specific details of the differences and similarities in cheeses ripened for 6 and 12 months. Which cheese is more suitable for which consumer group and why. Please refer directly to the purpose of the work, has the goal been achieved?
Reply 20. The conclusions have been rewritten according to the referee's suggestions.
- Please clearly indicate (in the abstract and conclusion) what is novel aspect and what new knowledge this research brings to science and technology. Please describe in a strong and blunt manner the validity of the research presented here.
Reply 21. The abstract and conclusions have been rewritten according to the referee's suggestions.
Round 2
Reviewer 3 Report
Comments and Suggestions for Authors
Dear authors
The paper, as you can see from my reviews, has problems in data analysis, both from a statistical and chemical point of view. I believe it can be improved and become interesting. That's why I spent my time to contribute for this task.

Author Response
Manuscript ID: foods-2697654 (2nd Reply)
Caciocavallo Podolico cheese, a Traditional Agri-food Product (TAP) of Basilicata region: nutritional, health, sensory properties and digital communication.
The authors thank the reviewer and have reported below the responses to the considerations made by the reviewer.
REV 1: As can be seen above, the authors analysed 3 cheeses aged 6 months (6M) and 3 cheeses aged 12 months (12M), from each of 5 producers. Therefore, 5×3×2=30 results are expected in total, divided by group (6Mand 12M) as this was the authors' main objective for this study. It would be useful if the number of replications carried out was added to this design (in some analyses it is said that there were two, in other analyses this is not declared).
Reply 1: All analytical determinations were performed in duplicate, and this was reported in the MM section of each determination, either at the end or within the description of the individual analytical method.
In MM Lines 135-137, this sentence was added: For each of the 5 producers, 3 cheeses were sampled for each ripening period. Therefore, the survey's experimental design involved sampling 30 cheeses (5 producers X 3 cheeses X 2 ripening periods).
At the end of paragraph 2.1. Line 143, was added this sentence: All analytical determination for each cheese sample was carried out in duplicate.
Rev3: “It was only justified to analyse normality by group, in each variable, justified to analyse normality by group, in each variable, and thus justify the subsequent use of parametric methods”
Reply 2: In fact, this was done, i.e., by analysing the within-group normality for each variable.
REV 3: Next, the authors chose to compare the two groups (6M and 12M) using the parametric method ANOVA. It is my opinion that the comparison of two groups requires the parametric Student’s t-test.
Reply 3: The authors accepted the reviewer's advice and analysed the data with Student's t-test. The SYSTAT statistical software only processes data that has been shown to be normally distributed using the Shapiro-Wilk test before applying the Student's t-test.
The tables show each group’s average value and SEM and the significance of the difference between the average to facilitate understanding.
In all tables, the average and SEM have been approximated to the first digit after the point.
REV 4: Finally, the authors are clear in saying that they consider that there are no differences if p > 0.1, that
there is a trend if p < 0.1 and that there are differences if p < 0.05. Therefore, as in most situations, p >
0.1 is found, the option should be to consider no differences between-groups and, consequently, speak
only of average values.
REPLY 4: The authors did not compare differences between groups in their discussion of results. They reported the absence of significant differences between groups and discussed average values with available literature for each parameter.
REV 5: The only analysis in terms of proteins was the kjeldahl method. This method only analyzes nitrogen.
For this reason, it cannot be used to discuss proteolysis. Nitrogen is neither lost nor gained during
drying, so the results of the analyzes could not have given a difference. If the authors wanted to
discuss proteolysis, which causes an increase in protein solubility, they should have homogenized the
sample in solution in the presence of trichloroacetic acid, which precipitates the protein, leaving
soluble protein segments in suspension (or another comparable method). This is a chemical error and
not a statistical one, which must be corrected.
REPLY 5: The authors added a reference to Fallico et al. [55] upon request by another reviewer during the first revision.
Based on the clarification provided by the reviewer, it has been decided to remove the two sentences mentioned earlier and retain only the sentence that states, "Fallico et al. [55] observed a negative correlation between moisture and protein content in Ragusano cheese, a pasta filata cheese made from raw milk."
REV 6: ……There are some things that may have an explanation: for example, the reduction of unsaturated fatty acids caused by oxidation reactions, or the reduction of vitamins due to loss of antioxidant potential throughout the ripening time; but it will be more difficult to understand the increase in cholesterol or polyphenols with curing time, or the increase in saturated fatty acids,or antioxidant capacity. And explanations must be found taking into consideration that the animals and the producers are the same
REPLY 6: The levels of unsaturated fatty acids, vitamins, saturated fatty acids, polyphenols, and antioxidant capacity are not significantly different between the two types of cheese.
While the hypothesis of a reduction in monounsaturated fatty acids following oxidation reactions and a loss of vitamin antioxidant potential may seem plausible, we cannot confirm it. The reason is that the same producers supply both the 6-month-old and 12-month-old cheeses, but the animals producing the cheeses are not the same. Since the samples were taken on the same day, as reported in the objective of the work, they could not have come from the same animals.
REV 7: Regarding artificial senses analysis, data from each instrument were submitted to Principal Component Analysis (PCA) for an unsupervised screening of the main differences between cheeses of two ripening periods. Then, the sensors with higher discriminatory power were selected, and ....
While I already mentioned in my previous review that when looking for differences the analysis to consider is discriminant analysis (DA), I understand that authors are more comfortable with principal component analysis (PCA). But in this case, it is necessary to understand that PCA reduces the data to components and that the components do not represent differences between groups but rather the main lines of variation. Thus, it would be better to exchange the "unsupervised screening of the main differences between cheeses from two ripening periods", through an "unsupervised screening of the main lines of variation, in the hope of highlighting any differences between ripening times.
It is important to see that, generally speaking, the graphs show no differences between ripening groups, but the existence of groups (probably related to producers) seem to exist, supporting my previous comments.
REPLY 7: Thanks for the important suggestions.
In 2.10 Statistical analyses section, lines 279-280 and lines 286-287, were added clarifications of the methodology
Although our intention was not necessarily to find differences, we understand that when describing them, it is better to use discriminant analysis to highlight them. Once the data fusion was carried out, we subjected the data set to DFA and replaced the PCA graph with that of the DFA in the paper (Figure 6). The validation score of 100 underlines the differences between the maturing times of the two groups of cheeses.
REV 8: ……..The main problem in typical cheeses is to reduce natural variations, through training and
advice. And if we want a well standardized cheese, then producers must come together in a
cooperative.
REPLY 8: Thanks to the reviewer for their suggestion. In addition to establishing a certification mark to protect the territory that identifies the product, we added a call to action for establishing a producers' cooperative in the conclusions.
REV 9: Problems persist in writing units, such as, for example, 2 grams instead of 2 g, two grams instead of 2g, 15 minutes instead of 15 min, 3 seconds instead of 3 s, and so on. This problem must be corrected.
Problems also persist with the design of the experimental procedure, with statistics, and explaining
results.
REPLY 9: Corrections have been made

Reviewer 4 Report
Comments and Suggestions for Authors
Thank you for improving your manuscript.
Author Response
We thank the reviewer for the time he dedicated to us